# The marginal cells of the *Caenorhabditis elegans* pharynx scavenge cholesterol and other hydrophobic small molecules

Muntasir Kamal[1,2], Houtan Moshiri[1,2], Lilia Magomedova[3], Duhyun Han[2,4], Ken C.Q. Nguyen[5], May Yeo[1,2], Jessica Knox[1,2], Rachel Bagg[1,2], Amy M. Won[2,6,7,8], Karolina Szlapa[1,2], Christopher M. Yip[2,6,7,8], Carolyn L. Cummins [3], David H. Hall [5] & Peter J. Roy[1,2,4]

The nematode *Caenorhabditis elegans* is a bacterivore filter feeder. Through the contraction of the worm's pharynx, a bacterial suspension is sucked into the pharynx's lumen. Excess liquid is then shunted out of the buccal cavity through ancillary channels made by surrounding marginal cells. We find that many worm-bioactive small molecules (*a.k.a.* wactives) accumulate inside of the marginal cells as crystals or globular spheres. Through screens for mutants that resist the lethality associated with one crystallizing wactive we identify a presumptive sphingomyelin-synthesis pathway that is necessary for crystal and sphere accumulation. We find that expression of sphingomyelin synthase 5 (SMS-5) in the marginal cells is not only sufficient for wactive accumulation but is also important for absorbing exogenous cholesterol, without which *C. elegans* cannot develop. We conclude that sphingomyelin-rich marginal cells act as a sink to scavenge important nutrients from filtered liquid that might otherwise be shunted back into the environment.

[1] Department of Molecular Genetics, University of Toronto, Toronto, ON M5S 1A8, Canada. [2] The Donnelly Centre for Cellular and Biomolecular Research, University of Toronto, Toronto, ON M5S 3E1, Canada. [3] Department of Pharmaceutical Sciences, Leslie Dan Faculty of Pharmacy, University of Toronto, 144 College Street, Toronto, ON M5S 3M2, Canada. [4] Department of Pharmacology and Toxicology, University of Toronto, Toronto, ON M5S 1A8, Canada. [5] Department of Neuroscience, Albert Einstein College of Medicine, New York, NY 10461, USA. [6] Institute of Biomaterials and Biomedical Engineering, University of Toronto, Toronto, ON M5S 3E1, Canada. [7] Department of Chemical Engineering and Applied Chemistry, University of Toronto, Toronto, ON M5S 3E1, Canada. [8] Department of Biochemistry, University of Toronto, Toronto, ON M5S 3E1, Canada. Correspondence and requests for materials should be addressed to P.J.R. (email: peter.roy@utoronto.ca)

The survival of life in the wild is dependent on enduring the boom and bust cycles of nutrient availability. This is especially true for the free-living nematode *Caenorhabditis elegans* that must survive extreme population bottlenecks because of limited nutrients[1]. Under ideal conditions, *C. elegans* will undergo embryogenesis and develop through four larval stages called L1–L4 to reach reproductive maturity in as little as 3 days[2]. Upon nutrient deprivation, overcrowding, or excessive temperature, the animal will enter an alternative stress-resistant long-lived dispersal state called dauer instead of L3[3]. Extensive sampling of nematodes in the wild reveals that *C. elegans* is most often found in the dauer state, unlike other free-living nematodes outside of the *Caenorhabditis* genus that are found to be actively proliferating[1]. Hence, *Caenorhabditis* seems particularly sensitive to stress, including nutrient deprivation.

*C. elegans* is a sterol auxotroph[4]. In the laboratory, *C. elegans*' food (*E. coli*) is supplemented with cholesterol, but other select fungal, plant, and animal sterols will also suffice[5]. The worm modifies the sterols for use in signaling and, unlike vertebrates, the sterols may be a structural component within the membranes of only a limited number of cells[4]. Depending on the extent and timing of sterol withdrawal, *C. elegans* will either enter the dauer state or fail to develop and perish[5]. A lack of select sterols, which are relatively insoluble and rare in the wild, is likely one factor that contributes to *Caenorhabditis* existing in the dauer state outside of the laboratory[6]. Mechanisms that maximize the absorption of these essential and scarce nutrients must therefore be essential to the overall success of the species.

*Caenorhabditis* is a filter-feeder[7,8]. Its feeding organ, called the pharynx, has a threefold symmetry with three muscles divided by three sets of marginal cells, all radially oriented with respect to the central lumen (see figures for schematics)[7]. The first step of the feeding cycle is that the radially aligned muscles of the pharynx contract, thereby opening the central lumen and creating negative pressure that sucks the bacterial suspension from the environment into the lumen. The pharynx then relaxes in an anterior to posterior wave, concentrating the suspended bacterial cells centrally within the lumen. Because of increasing pressure, the excess liquid escapes to the environment in a posterior-to-anterior manner via channels within the marginal cells[9]. The microbes are then macerated in the posterior pharynx and passed on to the neighboring intestine via peristaltic movements[7].

Here we report our discovery that select hydrophobic molecules visibly accumulate in the marginal cells of the *C. elegans* pharynx. No other tissue visibly accumulates hydrophobic small molecules. Through a forward genetic screen, we identified several components of a presumptive sphingomyelin (SM) synthesis pathway, at least one of which is specifically expressed in the pharynx, that are necessary for this accumulation. We find that the expression of the presumptive terminal enzyme of this pathway (sphingomyelin synthase 5 (SMS-5)) specifically in the marginal cells is sufficient for wactive accumulation. These observations led us to the finding that the marginal cells play an important role in the absorption of cholesterol, which is consistent with previous work showing that the pharynx accumulates sterols and other natural products[10,11]. Together, our observations indicate that the marginal cells scavenge nutrients from fluid that would otherwise be discarded by the worm. This nutrient-scavenging mechanism may be important in allowing *Caenorhabditis* to survive periods of limited nutrient availability.

## Results

### Wactives accumulate as crystals and spheres in the pharynx.

Observations from our previous drug screens revealed that animals incubated in some compounds have unusually dark pharynxes when visualized at ×100 magnification[12–14]. We investigated this phenomenon further by incubating synchronized first larval stage (L1) wild-type worms in 238 compounds from our wactive library at a concentration of 30 μM for 48 h (Fig. 1a). The wactive library is a collection of 627 compounds that we previously found to have bioactivity in *C. elegans*[12–14]. The 238 compounds that we surveyed are from plates 1 and 2 from the wactive library, which are enriched for potent bioactives, plus additional wactives that our group had working stocks of during the survey.

We visualized the worms at ×400 magnification using differential interference contrast (DIC), which allows clear visualization of unusual objects, and polarized light (birefringent) microscopy, which allows for the detection of objects containing molecules that are arrayed in a regular pattern (i.e., are crystalline)[15]. We found that 38 (16%) wactives accumulated as birefringent crystals and another 33 (14%) accumulated as non-birefringent globular spheres (henceforth referred to as spheres) (Fig. 1a–c). Of the 115 molecules that kill worms at a concentration of 30 μM, 66 (57%) form crystals or spheres in at least 25% of the animals (Fig. 1 in Source Data File). The only cells in which we found these unusual objects are within the corpus (anterior half) of the pharynx.

We investigated whether the crystals and spheres are a response of the worm to the wactives or whether the objects are likely to be the wactives themselves. We found that one of the crystallizing wactives, wact-469, fluoresces yellow-green (Supplementary Fig. 1), while no other crystallizing wactive that we investigated in detail fluoresces using our detection methods. We found that crystals that form in response to wact-469 fluoresce yellow-green, but crystals that form in response to other wactives like wact-190 do not fluoresce (Fig. 1b). Similarly, the wactive wact-43 fluoresces blue, and results in blue fluorescent spheres in the animal, whereas other non-fluorescent wactives like wact-209 do not produce fluorescent spheres (Fig. 1b). We conclude that the crystals and spheres are at least partly, if not entirely, composed of the respective wactive compound.

We investigated whether there are basic physicochemical features of the wactives that correlate with crystallization and sphere formation. We found that both crystal-forming and sphere-forming wactives are less hydrophilic (and have fewer hydrogen bond donors, which is related to hydrophilicity) on average than the wactives that do not generate unusual objects (Fig. 1d). This trend is consistent with the molecules precipitating out of solution when concentrated within the animal. One trend that may distinguish the crystal-forming compounds from those that form spheres is that the former have a greater topological polar surface area (PSA) on average (Fig. 1d). A key determinant of whether a membrane protein will crystallize in X-ray crystallographic studies is its topological PSA; more exposed polar groups increase the likelihood that a membrane protein will crystallize easily[16]. This raises the possibility that the crystallizing wactives are associating with the plasma membrane (PM) and coming out of solution in crystalline form due to their relatively large PSA.

To probe these trends further, we tested a set of ten newly purchased molecules whose physicochemical properties fall on one side of the trend toward crystallization and another set of ten newly purchased molecules whose physicochemical properties fall on the other side of the trend (away from object formation). Of the ten molecules with crystal-like physicochemical properties, two crystallized in the anterior pharynx. By contrast, none of the molecules with non-object-like physicochemical properties formed objects in the pharynx. That 20% of the molecules with crystal-like physicochemical properties form crystals is an

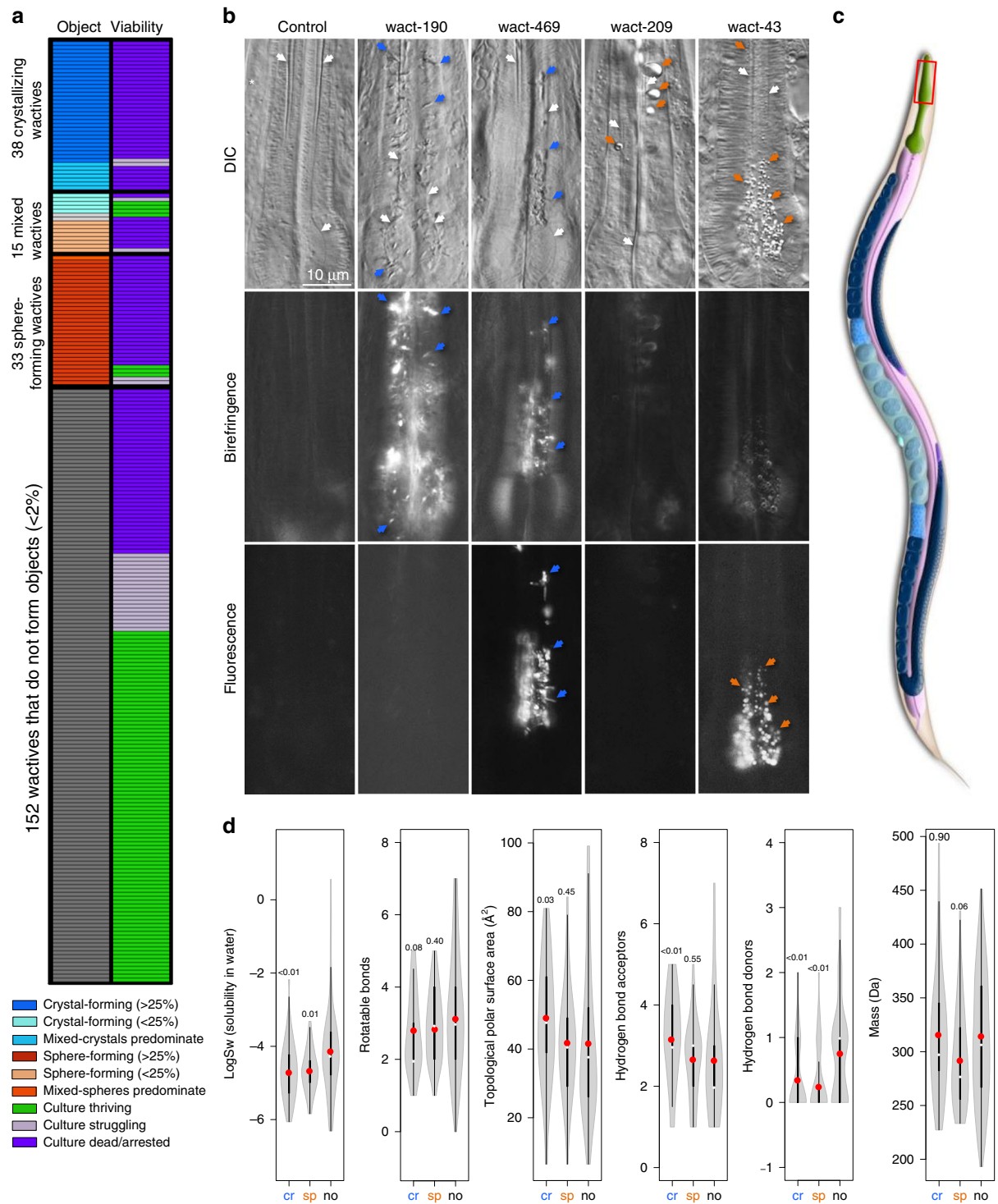

**Fig. 1** Wactives can form objects in the marginal cells of the anterior pharynx. **a** A chart showing our survey of the effects of 238 wactives on *C. elegans* L1 animals. Each row is a distinct wactive. The first column indicates whether a wactive accumulates as a crystal, sphere, or neither at a concentration of 30 μM. The second column indicates whether the wactive also kills/arrests worm development at a concentration of 30 μM. The percentages in the legend refer to the percentage of animals harboring the indicated objects. See "Methods" for details. **b** Examples of the accumulation of crystals and spheres in the anterior pharynx of young adults treated for 24 h to allow easy visualization of the channels. The first row shows differential interference contrast images; the second row shows whether the objects are birefringent (and therefore crystalline); and the third row shows whether the object fluoresces. White arrows indicate the channels, blue arrows indicate crystals, and orange arrows indicate spheres. The white asterisk indicates where the micrograph was spliced to ensure channel is in the correct focal plane. The scale is the same for all panels. **c** A schematic of *C. elegans* (courtesy of WormAtlas). The anterior pharynx is boxed. **d** Violin plots of the physical–chemical properties of molecules that crystallize (cr) (*n* = 38 distinct molecules), form spheres (sp) (*n* = 33 distinct molecules), or fail to form objects in the animal (no) (*n* = 152 distinct molecules). The gray shape represents all results and the thickness indicates how common that value is within the dataset; a white circle indicates the median; a red circle indicates the mean; the center thick line represents one standard of deviation; the thinner line represents the second standard of deviation. The number above each plot shows the significance of the difference compared to the "no-object" data using the nonparametric two-tailed Mann–Whitney *U* test. Source data for **a** and **d** are provided in the Source Data file

enrichment of 147-fold over what is expected by random chance (Supplementary Table 1).

**Wactive crystals may contribute to the death of *C. elegans*.** The majority of wactives that form crystals and spheres at 30 μM also induce developmental arrest and/or death of L1 larvae at that concentration (Fig. 1a). To better understand the relationship between object formation and lethality/arrest, we performed a dose–response analysis on a small subset of wactives that form crystals or spheres (Fig. 2a). We collected synchronized first larval stage animals (L1s) and inspected the pharynxes and other tissues at higher magnification after 48 h of incubation in the wactives. In parallel, we performed our standard 6-day viability assay[14]. We found a strong correlation between the presence of objects in the anterior pharynx and the associated lethality/arrest (an average $R^2$ of 0.964). We also asked whether the lethality associated with crystal-forming wactives can be avoided if the animals are removed from the wactive. We found that the lethal effect of the wactives can be prevented for most of the animals if they are removed from the wactive during the first 2 days of co-incubation with the molecule; thereafter, most animals do not recover from wactive exposure (Supplementary Fig. 2).

Many of the object-forming wactives are structurally diverse but have crystal or sphere formation in common (Fig. 2a). These observations raised the possibility that the objects themselves are contributing to the death or arrest of the animal. We reasoned that the crystals might perforate the membrane of the anterior pharynx and lead to death. We have no similar hypothesis about the spheres. We incubated the animals in the crystal-forming wactive wact-190 and sphere forming wact-34 and tested whether a membrane impermeable dye permeates the pharynx. We found that wact-190-exposed worms allow the permeation of Evans Blue dye[17] (Fig. 2b, c, e). By contrast, wact-34 failed to allow Evans Blue dye entry (Fig. 2d, e). These data are consistent with the idea that crystal formation may contribute to the death or arrest of the nematode by disrupting PM integrity and that spheres are simply a visible manifestation of the accumulation of select small molecules.

**The marginal cells act as a sink for select wactives.** To investigate the dynamics of crystal formation, we performed a time-course analysis of wact-190 crystal formation. We found that small birefringent objects can be seen as early as 30 min after incubation with wact-190 (Fig. 3a–c). By 48 h, the pharynx lumen of many animals is entirely filled with wact-190 crystals.

Early in the time course, wact-190 crystals accumulate near the channels of the anterior pharynx (Figs. 3a–c and 1b). The channels are formed by specialized cells called marginal cells and are part of the filtering mechanism by which the animal feeds on bacterial cells within its environment. The area of crystal and sphere formation, together with the animals' filter-feeding strategy, raised the possibility that crystals might be filtered from the media, get trapped in the channels, and seed the growth of larger crystals. Three observations argue against this. First, the birefringent crystalline objects that can been seen early in the time-course experiment are not found in the lumen of the channels but are instead associated with the cytoplasmic face of the basal lamina of the marginal cells (see the 30-min time point in Fig. 3). Second, crystals accumulate in the pharynx of young larvae that hatch within the parent's uterus and are not exposed to the external media (Supplementary Fig. 3). Third, 0 out of 17 molecules that precipitate in the media and fail to perturb the worm's development lead to crystal formation (Supplementary Fig. 4). Together, these observations argue against the idea that marginal cell-associated crystals are seeded from crystals in the

media. Instead, it is more likely that soluble small molecules accumulate in the tissue of the anterior pharynx, precipitate out of solution, and form crystals or spheres therein.

To examine the location of crystals in greater detail, we inspected the location of crystals using transmission electron microscopy. In cross-sections of wild-type animals incubated with wact-190, we found unusual structures that are consistent with the crystals we see with light microscopy. The unusual objects are located in association with the PM of the marginal cells and extend into the cytoplasm (Fig. 3d–g). In transverse sections of wild-type animals incubated with wact-190, we also observe crystal-like objects in the marginal cells. In addition, we see areas of clear material and/or separation of the PM from the luminal cuticle of the marginal cells (Supplementary Fig. 5). These observations are consistent with the idea that the marginal cells have unique properties that facilitate the absorption of small molecules with the physicochemical properties described above.

**SMS-5 functions cell-autonomously in the anterior pharynx.** To better understand the accumulation of crystals in the anterior pharynx, we performed a forward genetic screen to isolate mutants that resist the lethal effects of the crystal-forming wact-190. We reasoned that, if crystal formation is related to wact-190's lethality, mutants that resist its lethality would also resist crystal formation. We screened 1.3 million randomly mutagenized wild-type F2 genomes and isolated 46 mutants that resist the lethal effects of wact-190 (Fig. 4; Fig. 4 in Source Data File). A screen of 200,000 mutant F1 genomes did not yield wact-190-resistant mutants, suggesting that the resistant mutants isolated in the F2 screens are likely recessive.

Sequencing the genomes of the 46 mutants revealed 29 alleles of *pgp-14*, whose contribution to small molecule accumulation will be discussed elsewhere. Sequencing also revealed 12 strains that have mutations in 3 different components of a predicted de novo SM biosynthetic pathway (Fig. 5a). Sphingomyelin is one of the four major phospholipids that constitute the metazoan PM. Sphingomyelin and phosphatidylcholine are the major lipids in the outer leaflet of the PM, while phosphatidylserine and phosphatidyletherolamine are the major lipids within the inner leaflet[18,19].

Five of the wact-190-resistant strains have a mutation (including three early non-sense alleles) in *sms-5* (a.k.a. W07E6.3), which is a gene predicted to encode an integral membrane SMS and the terminal enzyme in the pathway[20] (Figs. 4 and 5a). SMS catalyzes the production of SM from phosphatidylcholine and ceramide on the outer leaflet of the PM[21]. Another five wact-190-resistant strains have a mutation (including two nonsense alleles) in *sptl-2*, which encodes a serine palmitoyltransferase. Two strains have a mutation (including one nonsense allele) in *ttm-5*, which is predicted to encode a sphingolipid delta(4)-desaturase. Both SPTL-2 and TTM-5 are predicted to function upstream of SMS-5 in the production of SM[21].

At each step of the SM synthesis pathway, there are multiple paralogs within the *C. elegans* genome that are predicted to perform the same biochemical role (Fig. 5a). We investigated independently derived deletion alleles of the paralogs including the genes we identified in our genetic screen. In every case, only the publicly available deletion allele of the respective mutant gene identified in our screen resisted the effects of wact-190 (Fig. 4). This suggests that either the specific expression pattern or the biochemical activity of the respective paralog identified in our screen is important for mediating sensitivity to wact-190. The *sms-5* null mutant resists wact-190's crystal formation (Fig. 5b–d), wact-190-associated developmental arrest/lethality (Fig. 5d), and the accumulation of wact-190 in the worm as revealed by mass

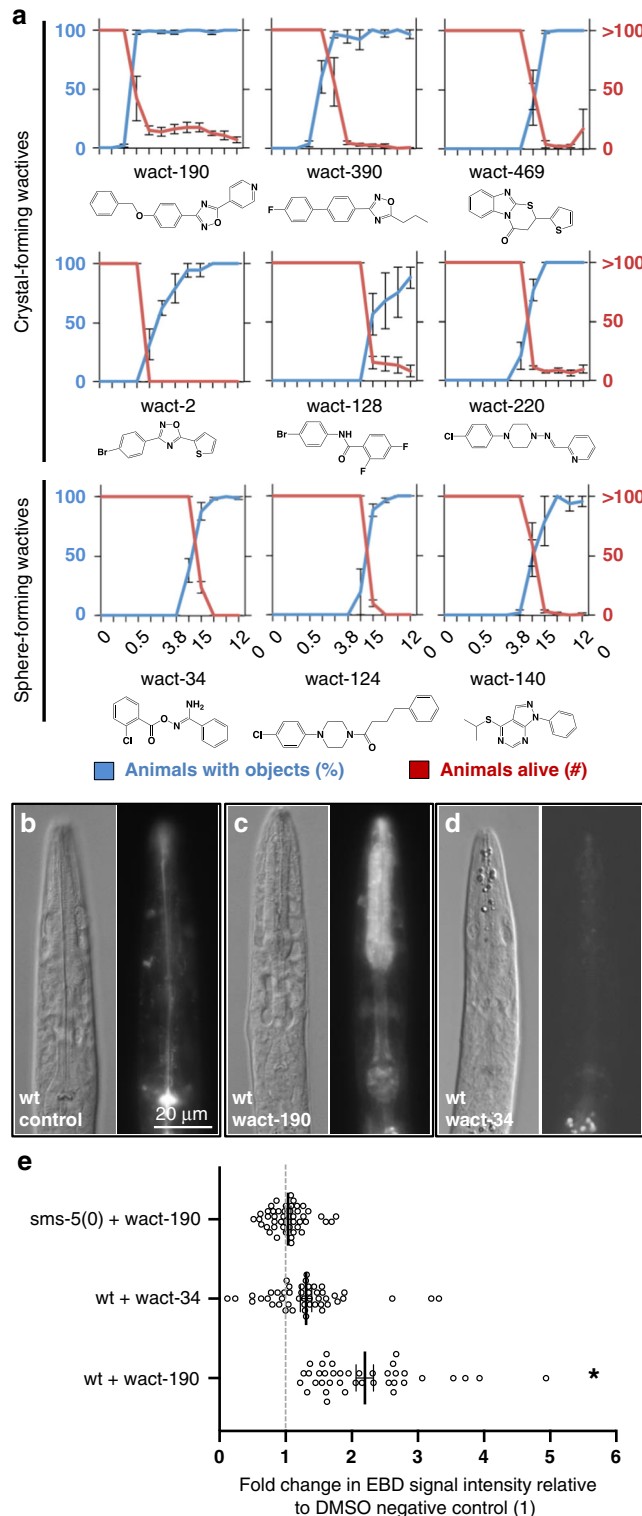

**Fig. 2** Wactive-induced lethality and object formation are correlated.
**a** Crystal and sphere formation in the pharynx was analyzed after a 48-h incubation of L1 larvae in the respective wactive and is indicated in blue. The lethality induced by the respective wactive, assayed after 50 L1s were incubated with the molecules for 6 days, is indicated in red. Animals in each well were counted if not over-grown with worms; otherwise, the well was scored as 100 animals. Low numbers of animals (see wact-190 or wact-128, for example) are invariably arrested L1s. The twofold serial dilution dose of the wactive, in micromolar, is indicated at the bottom of the three columns of graphs. The small molecule structure is illustrated below each graph. **b–e** An analysis of the ability of Evans Blue dye (EBD) to penetrate the anterior pharynx. L1-stage worms were incubated with control (1% dimethyl sulfoxide (DMSO) only) or 60 μM of the indicated compound for 24 h. Worms were then incubated in EBD for 4 h. **b–d** We see an enrichment of EBD signal in the anterior pharynx of worms treated with the crystal-forming wact-190 but not with the sphere-forming wact-34 or control. The scale is the same for all panels. For each pair of images, the differential interference contrast image is on the left, and the fluorescent image showing EBD signal is on the right. **e** Quantification of the EBD signal intensity of wact-190-treated worms relative to the DMSO negative control treated worms. Asterisk indicates a significant difference between the control and the wact-190-treated sample ($p < 0.05$) using Student's *T* test. Means and standard error are derived from $n = 3$ independent biological trials with at least six animals analyzed per sample per trial. Error bars represent standard error of the mean (SEM) for all graphs in this figure. Source data for **a** and **e** are provided in the Source Data file

clear enrichment in the anterior half (corpus) relative to the posterior half (isthmus and terminal bulb) (Supplementary Fig. 6). Animals mosaic for the transgene show SMS-5::YFP expression in the marginal cells (Fig. 5f, g).

Wild-type animals accumulate wact-190 crystals specifically in the anterior marginal cells. We therefore tested whether the expression of SMS-5 specifically in the marginal cells was sufficient to rescue the mutant's resistance to wact-190 crystal formation. We used a promoter (Supplementary Fig. 7) to express an SMS-5::FLAG::mCherry fusion protein (pPRHZ1138) specifically in the marginal cells of the anterior pharynx. For simplicity, we refer to this transgenic fusion protein as SMS-5(MC) henceforth. We found that expression of SMS-5(MC) can rescue the *sms-5* null phenotypes, including its resistance to both wact-190's lethality and crystal formation (Fig. 5d). Together, these data indicate that SMS-5 functions in the anterior pharynx to make the marginal cells especially sensitive to the accumulation of select small molecules.

**SMS-5 contributes to SM abundance in the pharynx.** Sphingomyelin facilitates a dense packing of lipids within a membrane[22]. Consequently, SM abundance is correlated to a resistance to solubilization by detergents[23,24]. We therefore tested whether *sms-5* mutants are more sensitive to detergents compared to wild-type animals. As predicted, we observed that the *sms-5* mutant is hypersensitive to detergents compared to wild-type controls. Furthermore, the expression of SMS-5 from the anterior marginal cells was sufficient to rescue this hypersensitivity to the detergents (Fig. 6a, b).

We next investigated the relative abundance of SM in the anterior pharynx using a protein probe called GFP-NT-Lysenin that binds to clustered SM (Supplementary Fig. 8)[25,26]. In wild-type fixed worms, we observed an enrichment of GFP-NT-Lysenin in tissues surrounding the pharynx, in the buccal cavity, and in the channels of the anterior marginal cells (Fig. 6c). The *sms-5* null mutant shows a significant decrease of fluorescent signal in the anterior pharynx (Fig. 6d, e). We infer that the

spectrometric analysis (Fig. 5e). The *sms-5* null mutant also resists Evans Blue dye penetration into the anterior pharynx when co-incubated with wact-190 (Fig. 2e).

We next investigated the tissues in which *sms-5* plays a role in small molecule accumulation. We used a fosmid-based reporter in which SMS-5 is tagged with yellow fluorescent protein (YFP) on its C-terminus (pPRHM1051). We observed SMS-5::YFP to be expressed in only two tissues: the spermatheca and the pharynx. In the pharynx, SMS-5::YFP is expressed in the muscle and marginal cells, with an enrichment at cell–cell junctions, and a

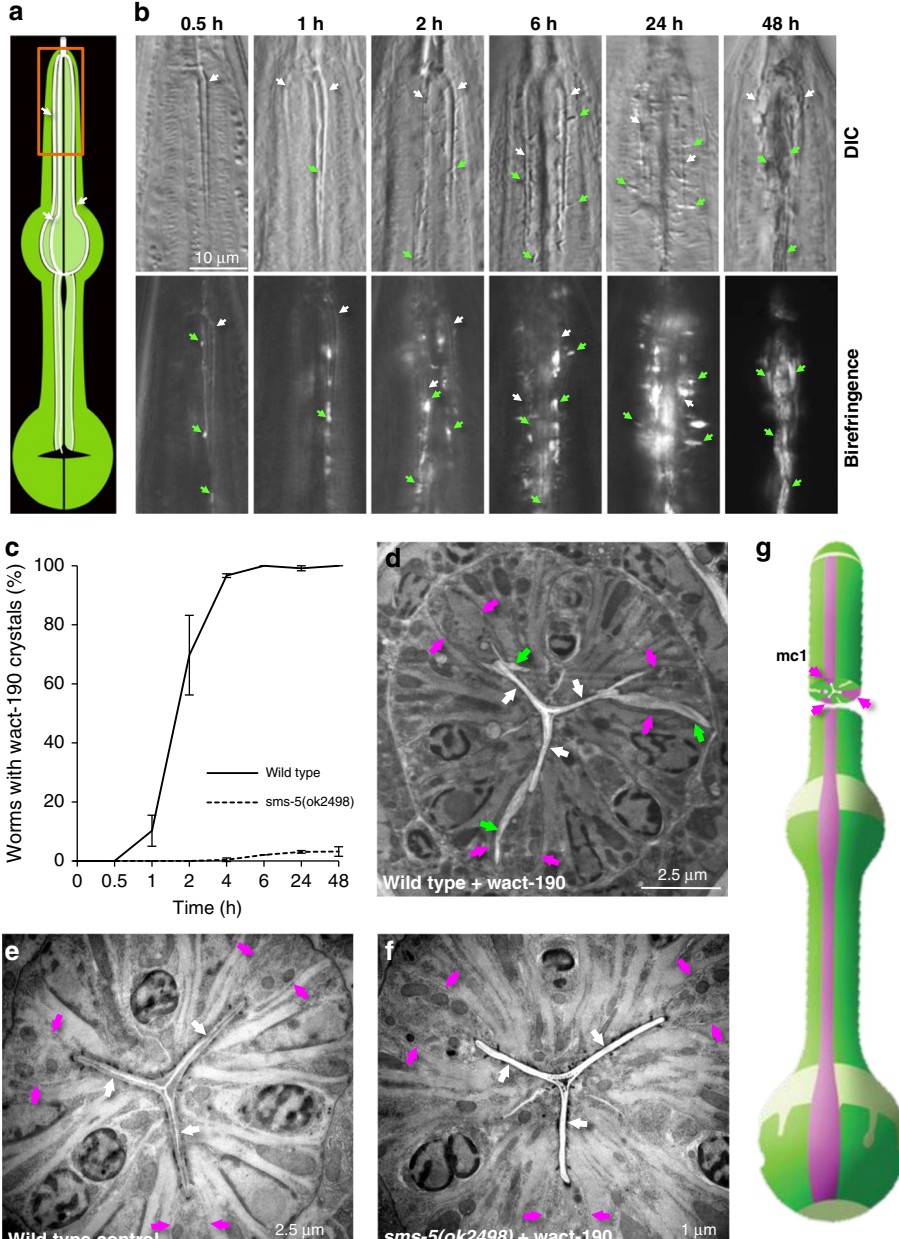

**Fig. 3** A microscopic analysis of wact-190 crystal formation. **a** A schematic of the pharynx with the three channels indicated with white arrows; the central lumen and grinder are in black. The procorpus of the anterior pharynx, which is shown in **b**, is boxed in orange. **b** Qualitative analysis of wact-190 crystal formation over time. Synchronized fourth-stage (L4) animals were grown in 60 µM of wact-190 in liquid culture for the time indicated at the top of each column. The top images show differential interference contrast micrographs; the corresponding bottom images are taken with polarized filters to visualize birefringence. Each pair of images shows the procorpus (anterior quarter) of the pharynx. The white arrows indicate channels; green arrows indicate select crystals. The scale is the same for all panels. **c** Quantitative analysis of wact-190 crystal formation over time. Synchronized first-stage (L1) worms of the indicated genotype were incubated in liquid with 60 µM wact-190 for the indicated time period and crystals were identified by their birefringence. Error bars represent standard error of the mean (SEM) and $n = 3$ independent biological samples. Source data are provided in the Source Data file. **d**–**f** Transmission electron micrographs (TEMs) of pharynx cross-sections of the indicated genotype and wactive treatment. The channels are indicated with white arrows; presumptive crystals with green arrows; marginal cell plasma membranes with fuchsia arrows. See Supplementary Fig. 5 for transverse sections. **g** A schematic of the pharynx illustrating the approximate area of the cross-sections imaged by TEM. The marginal cells are indicated in fuchsia. Image adapted from WormAtlas with permission

remaining signal is either background staining from the SM probe and/or SM that is produced by other SMSs[27]. We conclude that *sms-5* mutants produce less SM in the anterior pharynx compared to wild-type animals.

**The anterior pharynx is a filter for xenobiotic accumulation.** We investigated whether *sms-5* mutant animals are specifically

resistant to wact-190 or whether the mutant exhibits resistance to structurally diverse small molecules. To do this, we examined the viability of the wild type and the *sms-5(ok2498)* null mutant when incubated in 508 wactives at three different concentrations (7.5, 30, and 60 µM). In addition to wact-190, we found that the *sms-5* mutant is resistant to 23 other wactives in at least one of the concentrations tested (Fig. 7a). We refer to these 24 molecules as

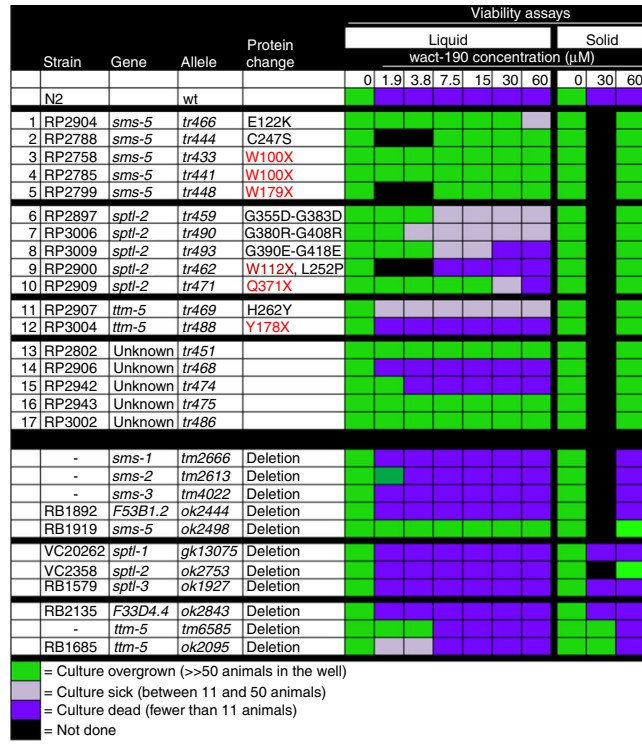

| | | | | Viability assays | | | | | | | | | |
|---|---|---|---|---|---|---|---|---|---|---|---|---|---|
| | | | | Liquid | | | | | | | Solid | | |
| | | | | wact-190 concentration (μM) | | | | | | | | | |
| | Strain | Gene | Allele | Protein change | 0 | 1.9 | 3.8 | 7.5 | 15 | 30 | 60 | 0 | 30 | 60 |
| | N2 | | wt | | | | | | | | | | | |
| 1 | RP2904 | sms-5 | tr466 | E122K | | | | | | | | | | |
| 2 | RP2788 | sms-5 | tr444 | C247S | | | | | | | | | | |
| 3 | RP2758 | sms-5 | tr433 | W100X | | | | | | | | | | |
| 4 | RP2785 | sms-5 | tr441 | W100X | | | | | | | | | | |
| 5 | RP2799 | sms-5 | tr448 | W179X | | | | | | | | | | |
| 6 | RP2897 | sptl-2 | tr459 | G355D-G383D | | | | | | | | | | |
| 7 | RP3006 | sptl-2 | tr490 | G380R-G408R | | | | | | | | | | |
| 8 | RP3009 | sptl-2 | tr493 | G390E-G418E | | | | | | | | | | |
| 9 | RP2900 | sptl-2 | tr462 | W112X, L252P | | | | | | | | | | |
| 10 | RP2909 | sptl-2 | tr471 | Q371X | | | | | | | | | | |
| 11 | RP2907 | ttm-5 | tr469 | H262Y | | | | | | | | | | |
| 12 | RP3004 | ttm-5 | tr488 | Y178X | | | | | | | | | | |
| 13 | RP2802 | Unknown | tr451 | | | | | | | | | | | |
| 14 | RP2906 | Unknown | tr468 | | | | | | | | | | | |
| 15 | RP2942 | Unknown | tr474 | | | | | | | | | | | |
| 16 | RP2943 | Unknown | tr475 | | | | | | | | | | | |
| 17 | RP3002 | Unknown | tr486 | | | | | | | | | | | |
| | - | sms-1 | tm2666 | Deletion | | | | | | | | | | |
| | - | sms-2 | tm2613 | Deletion | | | | | | | | | | |
| | - | sms-3 | tm4022 | Deletion | | | | | | | | | | |
| | RB1892 | F53B1.2 | ok2444 | Deletion | | | | | | | | | | |
| | RB1919 | sms-5 | ok2498 | Deletion | | | | | | | | | | |
| | VC20262 | sptl-1 | gk13075 | Deletion | | | | | | | | | | |
| | VC2358 | sptl-2 | ok2753 | Deletion | | | | | | | | | | |
| | RB1579 | sptl-3 | ok1927 | Deletion | | | | | | | | | | |
| | RB2135 | F33D4.4 | ok2843 | Deletion | | | | | | | | | | |
| | - | ttm-5 | tm6585 | Deletion | | | | | | | | | | |
| | RB1685 | ttm-5 | ok2095 | Deletion | | | | | | | | | | |

= Culture overgrown (>>50 animals in the well)
= Culture sick (between 11 and 50 animals)
= Culture dead (fewer than 11 animals)
= Not done

**Fig. 4** Mutant genes that resist the lethality induced by wact-190. Mutants listed in rows 1–17 were isolated in our forward genetic screen for those that resist the lethal/arrest phenotype of wact-190. In addition to these 17, we identified 29 other strains with a mutant allele of *pgp-14* that will be described elsewhere. Additional mutants beyond row 17 are deletion alleles obtained from either the *C. elegans Genetics Centre* or from Shohei Mitani. While all of the mutants isolated in our screen resist wact-190 on solid substrate, only some also resist wact-190 effects in liquid culture. *F53B1.2* is a paralog of the *sms* genes, and *F33D4.4* is a paralog of *ttm-5*. Viability assays were done with at least four replicates (see "Methods" for details). Residue numbers for both isoforms of *sptl-2* are shown. Protein changes noted in red indicate a presumptive null allele; "X" indicates an early nonsense codon. For five wact-190-resistant strains (rows 13–17), the mutant gene responsible for wact-190 resistance has not been identified and are not discussed further. Source data and homozygous mutation information for the relevant strains are provided in the Source Data file

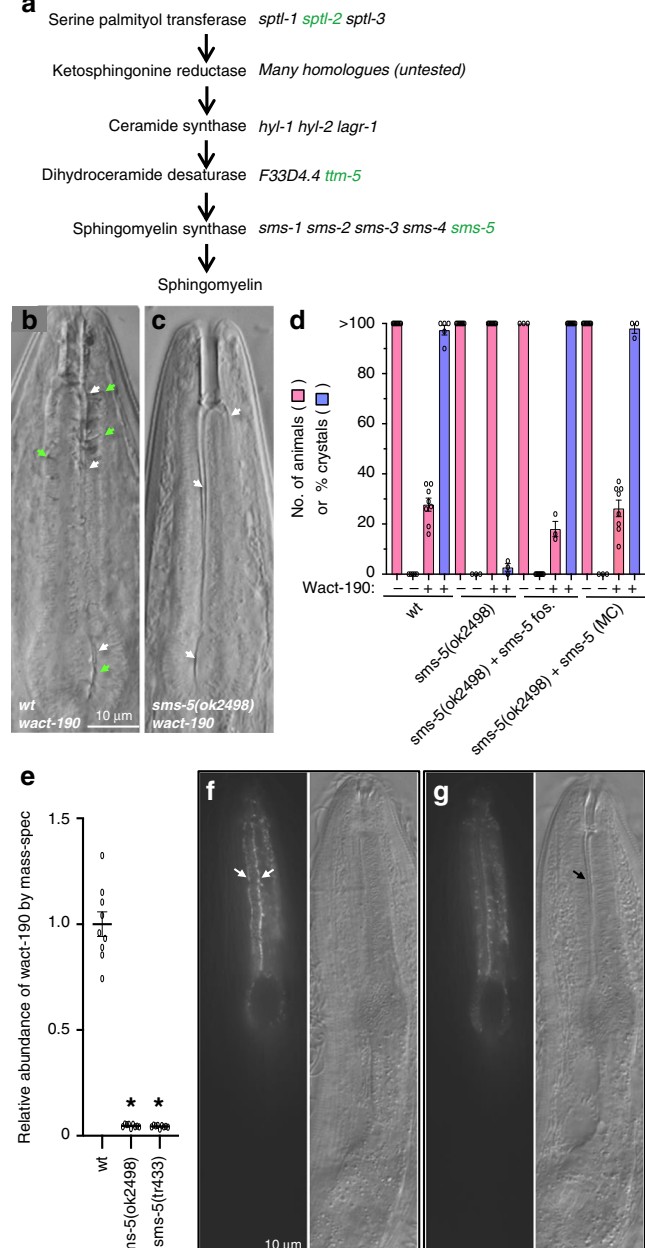

resistant wactives, the majority of which are structurally distinct. Detailed analyses with two resistant wactives show that SMS-5 expression specifically in the anterior marginal cells (SMS-5(MC)) rescues this resistance (Fig. 7b, c). Like wact-190, we measured the accumulation of two other resistant wactives by mass spectrometry and found decreased accumulation in the tissue of the *sms-5* null mutant relative to wild-type animals (Fig. 7f).

We were surprised to find that 169 (33%) of the wactives killed the mutants at a concentration that had little-to-no effect on wild type (Fig. 7a). Because the *sms-5* mutants are hypersensitive to these 169 molecules, we refer to these as hypersensitive molecules. Detailed analyses with two hypersensitive molecules show that SMS-5 expression specifically in the anterior marginal cells (SMS-5(MC)) can rescue hypersensitivity (Fig. 7d, e). In contrast to the decreased accumulation of resistant molecules in the *sms-5* mutant, we examined the accumulation of three hypersensitive molecules and found increased accumulation in the *sms-5* mutant relative to wild type (Fig. 7f).

We investigated the physical–chemical properties of the resistant and hypersensitive wactives. We found that the resistant molecules have a larger mass, have a greater topological PSA, are

significantly less soluble in water, and have more hydrogen bond acceptors, than the molecules that kill both the wild type and the *sms-5* mutant (Fig. 7g). The hypersensitive molecules have nearly the opposite set of properties: they are of lower molecular weight, have a smaller PSA, and have fewer hydrogen bond acceptors and donors than control molecules that fail to kill either the wild type or the *sms-5* mutant (Fig. 7g). We infer that SMS-5 contributes to a marginal cell barrier that normally allows the accumulation of large hydrophobic molecules but limits the accumulation of relatively smaller and less hydrophobic molecules.

**Some crystallizing wactives kill through multiple mechanisms.** Several lines of evidence are consistent with the idea that wact-190 crystals induced developmental arrest and/or lethality. First, the concentration at which crystals form is coincident with the concentration that kills worms (Fig. 2a). Second, the PM is

**Fig. 5** Mutations in a predicted sphingomyelin synthesis pathway confer resistance to wact-190. **a** A predicted sphingomyelin synthesis pathway. Whole-genome sequence analysis indicated that mutations in three genes predicted to play a role in sphingomyelin synthesis (*sptl-2*, *ttm-5*, and *sms-5*) are sufficient to confer resistance to wact-190 (see Fig. 4). Independently derived deletion alleles were tested for each of these genes (in green), together with their respective paralogs (in black), and only the deletions of *sptl-2*, *ttm-5*, and *sms-5* confer resistance (see Fig. 4). **b**, **c** 60 µM wact-190 forms crystals in wild-type worms but not in the *sms-5* (ok2498) mutant. Animals were incubated in the small molecule for 24 h as L4-staged animals. Anterior is up; white and green arrows indicate channels and crystals, respectively. **d** Quantification of wact-190's effects on viability and crystal formation. The *sms-5*-containing fosmid (WRM0626dC03) is harbored as an extra-chromosomal array. The construct expressing SMS-5:: FLAG::mCherry from an anterior marginal cell-specific promoter (SMS-5 (MC)) (see Supplementary Fig. 7) is harbored by the transgenic array (trIs104). Viability is quantified with n = 8 independent experiments; crystal counts were done with n = 3 independent experiments. In samples where viability is indicated as <40, the respective wells have only the original young larvae that are arrested or dead. **e** Accumulation of the indicated small molecule in *sms-5* mutants relative to wild-type animals as measured by mass spectrometry. n = 3 independent experiments. Asterisk (*) indicates a significant difference (p < 0.001) using a Student's T-test. **f**, **g** Two focal planes of an adult mosaic for the extra-chromosomal (Ex) array harboring pPRHM1051 (a construct with the SMS-5 fosmid WRM0626dC03 with yellow fluorescent protein (YFP) coding sequence inserted in frame at the C-terminus of SMS-5). The YFP-fusion protein is localized to the anterior marginal cells and is enriched basally at the borders with adjacent muscle cells (white arrows). Supplementary Fig. 6 shows the non-mosaic expression pattern of SMS-5. For each pair of images, fluorescent image is on the left and the differential interference contrast image is on the right. The scale is the same for all corresponding panels. Error bars represent standard error of the mean (SEM) for all graphs in this figure. Source data for **d** and **e** are provided in the Source Data file

disrupted in the same cells in which crystals form (Figs. 1 and 2). Third, *sms-5* mutants coincidentally resist the crystal formation and lethality induced by wact-190 (Figs. 5d and 7b, c). Fourth, the restoration of SMS-5 activity specifically in the cells in which crystals normally form restores both crystal formation and lethality to the *sms-5* mutant (Figs. 5d and 7b, c).

We therefore expected that all of the crystal-forming compounds would behave like wact-190. That is, we expected that all crystal-forming compounds would: (i) fail to form crystals in the *sms-5* mutant, and coincidentally, (ii) fail to kill the *sms-5* mutant. However, we were surprised to find that 53% of the crystallizing wactives that kill the wild type remain effective at killing the *sms-5* mutant (Fig. 8a).

To investigate the discordance of the ability to kill the *sms-5* mutant among crystal-forming wactives, we investigated a handful of crystal-forming compounds in detail for their ability to kill and form crystals in the *sms-5* mutant. In all cases, we found that the *sms-5* mutant resists crystal formation, but half of these wactives are still able to kill the mutant (Fig. 8b). Given that several lines of evidence indicate that crystals can kill larvae, these data suggest that some crystal-forming wactives may kill through both crystal formation and through crystal-independent mechanisms.

We also performed similar dose–response analyses for a handful of sphere-forming compounds. In all cases examined, *sms-5* mutants remained sensitive to these wactives despite a lack of sphere formation (Fig. 8b). This suggests that the sphere formation itself is simply a visible marker of compound accumulation and is not directly related to how the sphere-forming wactives kill the animals.

**sms-5 mutants are hypersensitive to the loss of cholesterol.** *C. elegans* is a cholesterol auxotroph, requiring as little as 2.5 ng/mL to survive[4]. CHUP-1 (initially called CUP-1) is a key transmembrane protein involved in cholesterol absorption in *C. elegans*[28]. Consequently, *chup-1* mutants are hypersensitive to cholesterol-limited conditions[28]. CHUP-1 is expressed in the posterior pharynx but is more prominently expressed in the intestine[28]. Despite *C. elegans*' reliance on mature sterols for development[4], *chup-1* null mutants are viable, suggesting that additional mechanisms of cholesterol uptake exist[28].

Given SMS-5's role in the accumulation of hydrophobic small molecules, we asked whether SMS-5 might function in parallel with CHUP-1 to help absorb cholesterol, which is also a small hydrophobic molecule. Wild-type hatchling parental (P0) animals placed on cholesterol-limited plates are themselves able to grow to adulthood, likely because of maternally contributed cholesterol stores[29]. However, these P0 animals produce fewer F1 progeny and these F1s are slow to develop to the L4 stage (Fig. 9a, first column). We tested how the *sms-5* null mutant behaves on cholesterol-limited plates. Like the *chup-1* null control, *sms-5* null animals had difficulty growing on cholesterol-limited conditions (Fig. 9a; Supplementary Fig. 9). The growth deficit of the *sms-5* mutant is rescued by the expression of SMS-5 specifically in the anterior marginal cells. Furthermore, the *chup-1; sms-5* double mutant grows worse on cholesterol-limited conditions than either single mutant (Fig. 9a; Supplementary Fig. 9). This double mutant analysis indicates that SMS-5 may function in parallel to CHUP-1 in the absorption of cholesterol.

We also investigated whether the *sms-5* mutant is deficient in cholesterol absorption by examining the uptake of a fluorescent analog called NBD-cholesterol, which was previously used to characterize *chup-1*'s role in cholesterol uptake[28]. Similar to previous results, we observed an obvious decrease in NBD-cholesterol accumulation in the *chup-1* mutants relative to wild-type control after synchronized L1s were incubated in NBD-cholesterol for 6 days (Fig. 9b–d; Supplementary Fig. 10). We observed a similar decrease in NBD-cholesterol accumulation in the *sms-5* mutant (Fig. 9e), which could be rescued by expressing SMS-5 specifically in the anterior marginal cells (Fig. 9f). Together, the data indicate that SMS-5 contributes to cholesterol absorption.

## Discussion

Here we show that the marginal cells within the anterior pharynx have distinct properties that facilitate unique interactions with exogenous small molecules. We can speculate why these marginal cells play a special role in the animal by first considering that nematodes are cholesterol auxotrophs and that free cholesterol is relatively rare within the environment[1,4]. The marginal cells form the channels through which excess ingested fluids are expelled from the animal[7,9]. Hence, the marginal cells may function as a salvage system to ensure that scarce nutrients are not again lost to the environment. This model is consistent with several observations: First, exogenous small molecules visibly accumulate almost exclusively in the cells that line the anterior channels. Second, the pharynx is the first tissue to accumulate a fluorescent analog of cholesterol in time-course analyses[30]. Third, the anterior pharynx accumulates obviously higher levels of cholesterol compared to the posterior pharynx[10]. Finally, absorption of these small molecules is SMS-5 dependent and SM-rich membrane has been repeatedly shown to act as a sink for cholesterol[24].

How SMS-5 facilitates the absorption of hydrophobic molecules is not clear. We know that sphingomyelin incorporation into a lipid bilayer alters the physicochemical properties of that bilayer (reviewed in refs. [31,32]). In other animals, SM is tightly

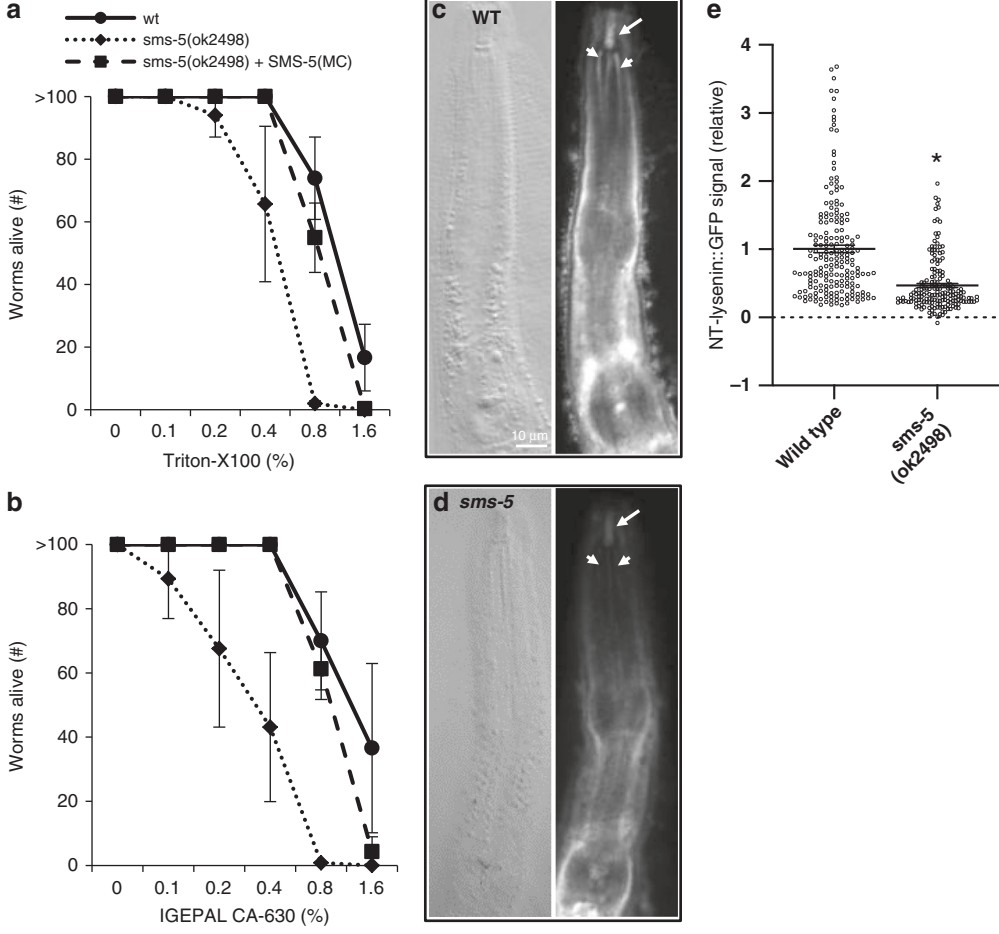

**Fig. 6** *sms-5* mutants have less sphingomyelin in the anterior pharynx. **a**, **b** The *sms-5* deletion allele is hypersensitive to the lethality induced by detergents triton-X 100 and IGEPAL CA-630, shown as a v/v percentage of the liquid media. In both cases, differences are significant ($p < 0.05$) at a detergent concentration of 0.8% (w/v) using a Student's *T* test. SMS-5(MC) represents marginal-cell-specific expression of SMS-5::FLAG::mCherry from the *trIs104* integrated transgenic array. Three biological repeats were done with four technical repeats done during each biological repeat. **c**, **d** Wild-type and *sms-5* (ok2498) mutant animals are fixed and stained with the GFP-NT-Lysenin protein probe that recognizes clustered sphingomyelin. White arrows indicate buccal cavity and white arrowheads indicate anterior channels. For each pair of images, the differential interference contrast image is on the left, and the fluorescent image is on the right. The scale is the same for all panels. **e** Quantification of the relative green fluorescent protein (GFP) signal from more than five biological replicates of GFP-NT-Lysenin staining (see "Methods" for details). Asterisk (*) indicates a significant difference ($p = 0.024$) using a Student's *T* test. $n = 7$ independent trials with at least 10 worms analyzed during each trial. Error bars represent standard error of the mean (SEM) for all graphs in this figure. Source data for **a**, **b**, and **e** are provided in the Source Data file

associated with cholesterol within the PM and creates subdomains of densely packed regions within the lipid bilayer[32]. In addition, sphingolipids and cholesterol have been shown to act as molecular traps for one another[33,34], which is consistent with the model that the SM-rich marginal cells of *C. elegans* can act as a sink for cholesterol and other hydrophobic molecules. Notably, a reduction in SM abundance decreases the packing density of lipids[23,24] and likely makes it easier for lower-molecular-weight molecules to pass through the PM barrier (Fig. 10). Hence, the biological properties of SM are consistent with our findings that SMS-5 activity leads to the accumulation of larger hydrophobic wactives while creating a barrier to smaller wactives.

Our observations suggest a potential paradox: *C. elegans* has evolved a sterol-salvaging system as part of its filter-feeding strategy despite simultaneously creating a vulnerability to xenobiotic crystal formation. However, it is unlikely that high concentrations of hydrophobic xenobiotics are commonplace in nature. Death by crystal formation may therefore be rare beyond the laboratory. Hence, a cholesterol-salvaging system may inadvertently facilitate the accumulation of hydrophobic xenobiotics

but be of little evolutionary consequence because crystal-forming concentrations of xenobiotics may rarely be encountered in the wild.

Our discovery that small molecules can kill worms through crystallization has important implications for drug screens using *C. elegans* as a model system. Crystallizing wactives are varied in structure and potency, but all have the potential to kill or arrest young larvae through perforation of the PM. For many of these crystallizing wactives, both crystallization and its associated lethality is dependent on an intact SMS-5 SM synthesis pathway (Fig. 10). This class of wactives likely has no other mechanism of killing aside from crystal formation. Our unpublished observations suggest that wactives that kill *C. elegans* through crystal formation alone do not have wide-ranging utility against diverse parasitic nematodes. Hence, our future screens for nematicides using *C. elegans* as a screening platform will include a step to determine whether the wactive's bioactivity is SMS-5 dependent.

A second class of crystallizing wactives are those whose lethality is not dependent on SMS-5. In an *sms-5* mutant background, the crystallization of this class of wactives is abolished, but their

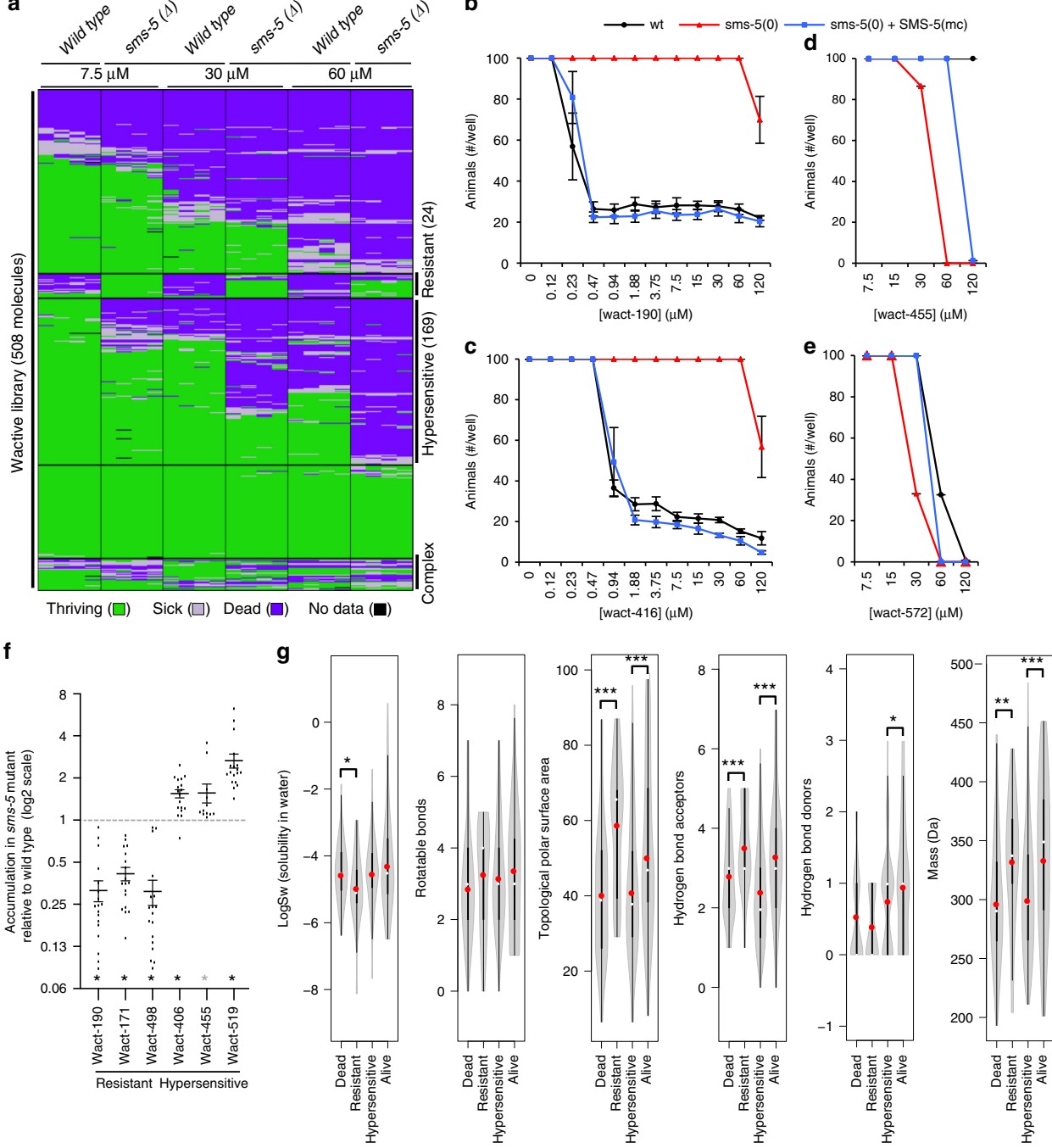

**Fig. 7** *sms-5* mutants have altered sensitivity to small molecules. **a** L1 viability assays of animals of the indicated genotype (top) grown in liquid culture in quadruplicate in 508 different wactive small molecules at 7.5, 30, or 60 μM. Wells were inspected after 6 days of growth; wells that had ≥50 animals are shown in green, wells that have between 11 and 50 animals are shown in yellow, and wells with ≤11 animals are shown in red. The resulting population growth of each of the four replicates is shown in each column. Each row corresponds to a distinct wactive molecule. The data are clustered along the *y* axis. The 24 wactives with reduced potency in the mutants are referred to as resistant wactives, and the 169 wactives with increased potency in the mutants are referred to as hypersensitive wactives. **b–e** Dose–response analyses of *C. elegans* population growth with two resistant molecules (wact-190 and wact-416) and two hypersensitive molecules (wact-455 and wact-572). The *sms-5(ok2498)* deletion mutant ("*sms-5(0)*") was used in this analysis, along with the rescuing *sms-5* transgene whereby SMS-5::FLAG::mCherry (SMS-5(MC)) is expressed exclusively in the anterior marginal cells. Resulting wells with <50 animals in the wact-190 and wact-416 conditions were invariably arrested or dead L1s. Viability counts were done with n = 8 independent experiments; crystal counts were done with n = 3 independent experiments. **f** Accumulation of the indicated small molecule in the *sms-5(ok2498)* mutant relative to wild-type L1 animals as measured by mass spectrometry. Standard error of the mean is shown for graphs **b–f**. n = 6 independent experiments for wact-190, wact-171, wact-498, wact-406, and wact-519; n = 4 for wact-455. **g** Violin plots comparing the physical–chemical properties of molecules that are classified as lethal in all conditions (dead), resistant, hypersensitive, or not obviously bioactive (alive) at 30 μM. See the legend of Fig. 1 for the description of the violin plots. One, two, and three asterisks indicate a significant difference (*p* < 0.05, *p* < 0.01, and *p* < 0.001, respectively) between the indicated datasets using Student's *T* test. Source data are provided in the Source Data file

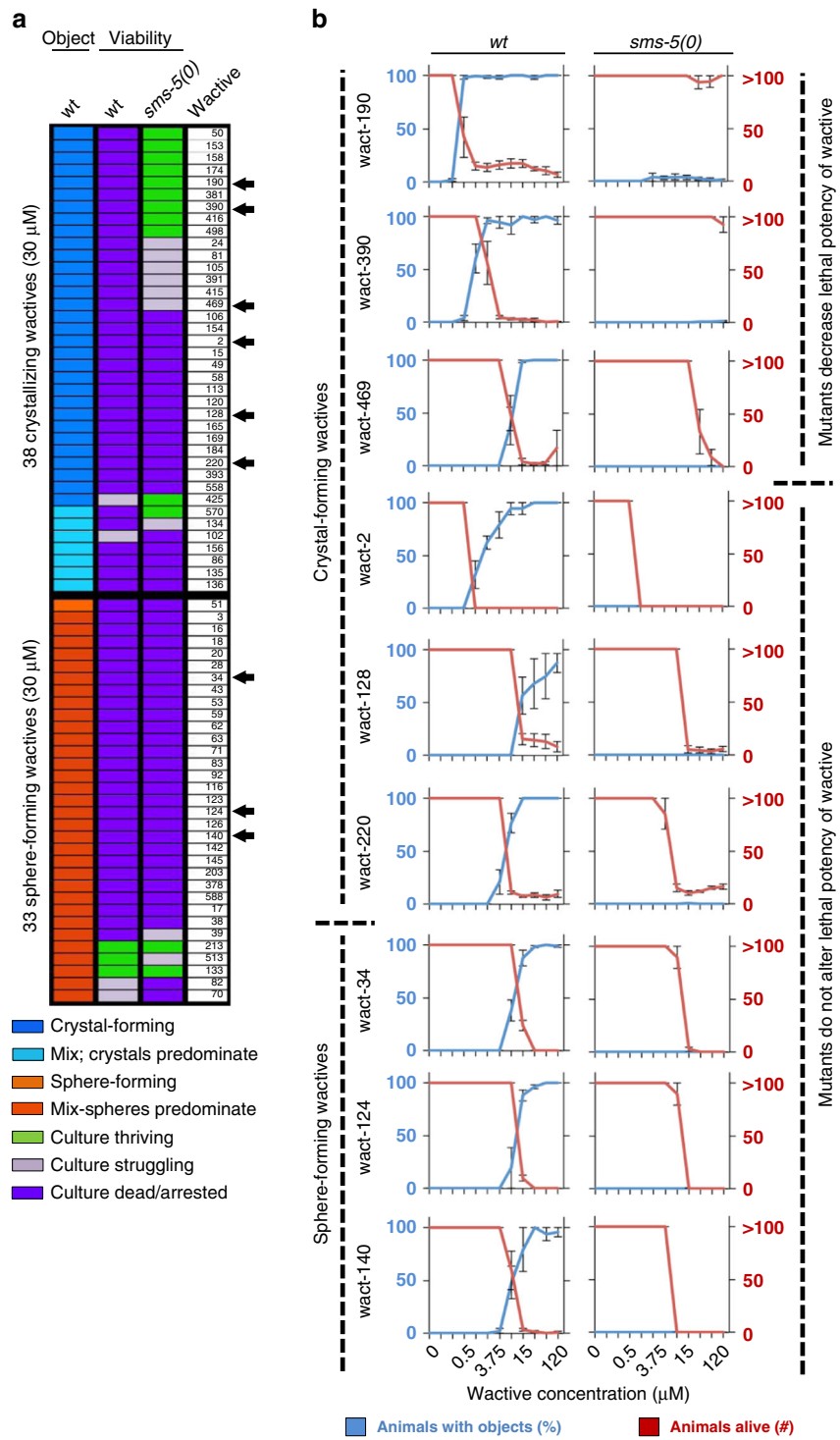

**Fig. 8** A comparison of the lethality and object-forming capability of wactives in animals lacking SMS-5. **a** The behavior of molecules that form objects in at least 25% of the population are analyzed with respect to their lethal potency in wild-type and the *sms-5(ok2498)* deletion mutant. The wactive number is shown on the right and black arrowheads indicate the wactives analyzed in **b**. **b** Dose–response analyses for select crystal and sphere-forming wactives in the indicated genetic background (indicated at the top of the two columns of graphs). Each of the wactives is analyzed over a twofold dilution series of concentrations (indicated at the bottom of the three columns of graphs). Wactive molecules are indicated on the left. The standard error of the mean is shown. Source data are provided in the Source Data file

lethality persists. This suggests that this class of crystallizing wactives has at least two mechanisms of killing— one that is crystal dependent, and one that is crystal independent. Knowing that this second class of wactives exist has important ramifications for how we investigate the mechanism of action of bioactive

molecules. Typically, our first approach toward characterizing a wactive's mechanism of action is to screen for *C. elegans* mutants that resist the molecule's lethal effects. When successful, the mutant gene that confers resistance provides insight into the molecule's mechanism of action[12–14,35]. However, with a

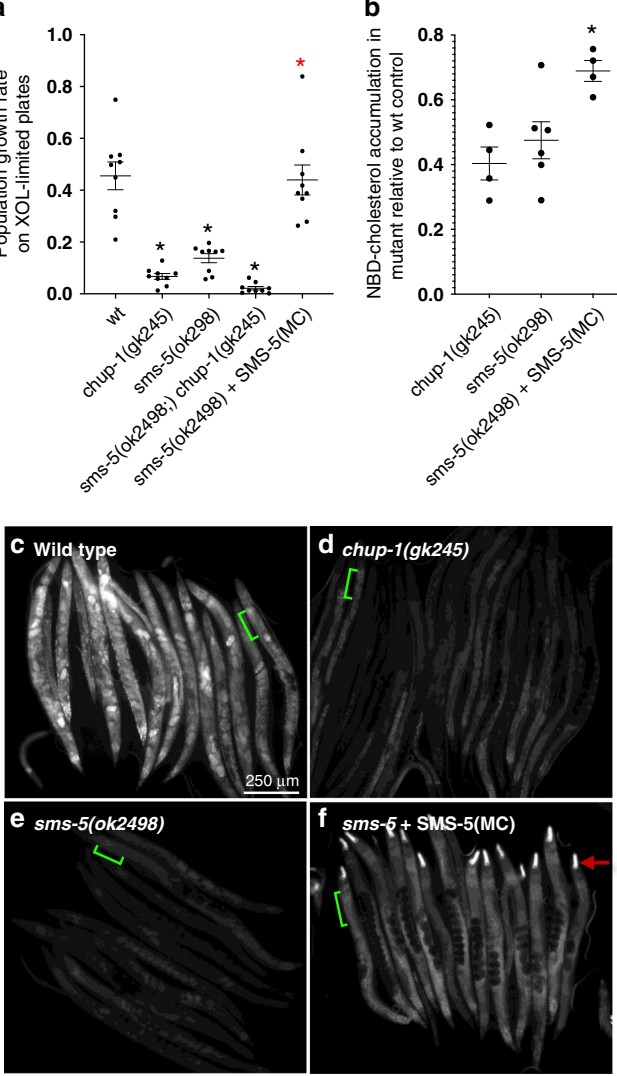

**Fig. 9** SMS-5 is required for development in cholesterol-limited conditions. **a** The growth rate of animals of the indicated genotype grown on cholesterol (xol)-limited plates (containing 50 ng/mL xol) is reported relative to the same strains grown on plates with standard concentrations of cholesterol (5000 ng/mL xol) ($y$ axis). See "Methods" for additional details. Asterisks indicate significance ($p < 0.001$) relative to the wild-type (wt) control; the red asterisks indicate significance ($p < 0.05$) relative to the *sms-5(ok2498)* deletion mutant. The double mutant has a more severe growth defect relative to either single mutant ($p = 0.06$). $n = 10$ independent biological trials for the control (wt); $n = 3$ independent biological trials for all experimental genotypes. **b** Accumulation of NBD-cholesterol signal of the indicated mutants relative to wild-type controls. Black asterisks indicate significance ($p < 0.01$) relative to the wild-type control; red asterisk indicates significance ($p < 0.05$) relative to the *sms-5(ok2498)* deletion mutant. $n = 6$ independent biological trials for wt and the *sms-5* mutant; $n = 3$ independent biological trials for the other two strains. In both **a** and **b**, standard error of the mean is shown and significance is measured using Student's $T$ test. Source data are provided in the Source Data file. **c–f** Images of multiple animals after the indicated strains were incubated in NBD-cholesterol for 6 days. The scale is the same for all images. Green lines exemplify the area used in each animal to calculate signal in **b**. Red arrow indicates the expression of yellow fluorescent protein that is restricted to the anterior pharynx that is used as a marker to follow the rescuing transgene *trIs104*. In **a**, **b**, and **f**, SMS-5(MC) represents marginal-cell-specific expression of SMS-5 from the *trIs104* integrated transgenic array. Supplementary Fig. 9 shows that, in the absence of NBD-cholesterol, each strain auto-fluoresces in the green fluorescent protein channel to the same extent

molecule that kills via two distinct mechanisms, any worm with a mutation that confers resistance to one mechanism will likely be killed by the second mechanism. Isolating a mutant that resists both mechanisms of action would be rare, occurring roughly once in every four million mutant genomes if loss-of-function mutations were sufficient to confer resistance to each lethal mechanism (and even rarer if resistance is conferred by gain of function only). Knowing that a bioactive molecule behaves in this way may facilitate the characterization of its mechanism of action by screening for resistant mutants in an *sms-5* null background wherein crystal formation is abolished.

Investigating the sensitivity of the *sms-5* mutant to the entire wactive library revealed that this mutant is at least twofold more sensitive than wild type to more than a third of the small molecules surveyed. Detailed analysis with a small subset of these hypersensitive molecules indicates that restoring SMS-5 function to only the marginal cells is sufficient to confer normal sensitivity to these wactives. These observations raise several important points. First, it raises the possibility that the tissue of the anterior pharynx is either the target of many bioactive molecules and/or serves as the conduit through which many bioactive molecules traverse en route to their targets in other tissues. Intriguingly, there are several sphere-generating wactives that fail to generate spheres in the mutant background but simultaneously increase in potency in the mutant background. This suggests that, without SMS-5 in the pharynx, the sphere-generating wactives cannot be contained within the pharynx and are better able to disrupt targets throughout the animal. Second, the *sms-5* mutant becomes a tool with which we can increase the potency of many small molecules. Given that the *sms-5* deletion mutant is otherwise healthy, it is a useful sensitized background for future small molecule screens and other chemical genetic experiments.

## Methods

**Developmental synchronization of *C. elegans* population**. To obtain synchronized populations of L1s, gravid adults were washed off the plates with M9 buffer, centrifuged at $800 \times g$ to remove supernatant, followed by additional washes in M9 buffer until all bacteria were removed after which worms were collected as a 1.5-mL pellet in a 15-mL conical tube[36]. Next, 1 mL of 10% hypochlorite solution (Sigma) was added to the tube followed by 2.5 mL of 1 M sodium hydroxide solution and 1 mL double-distilled water, and the mixture was incubated on a nutator for 5 min. With 1.5 min remaining for incubation, the tube was vortexed for 10 s with two 5-s pulses after which the harvested eggs were washed 4 times with 12 mL M9 buffer, with vortexing after every addition of M9 buffer. After the final wash, the tube was incubated overnight on a nutator at 20 °C to allow egg-hatching and checked the next day for synchronized L1s. To obtain other synchronized stages, the synchronize L1s are plated on solid agar substrate with *E. coli* food and allowed to progress to the desired stage before processing.

**Crystal and sphere analyses**. Synchronized L1s were added to wells of a 96-well plate (50 L1s/well) containing liquid Nematode Growth Medium (NGM), *E. coli* (HB101) as food source (OD$_{600}$ = 1.2), and 30 μM of 240 wactives in duplicate wells. Forty-eight hours later, worms were transferred to Eppendorf tubes and washed once with fresh M9 and spun down at $1800 \times g$ for 1 min to form a tight worm pellet of ~10 μL. Then 5 μL of 50 mM levamisole (to a final concentration of ~16.7 mM) was added to paralyze worms. Worms were then mounted on a 2% agarose pad on a glass slide. Live worms were then observed for the presence of birefringent crystals or non-birefringent spheres in the pharynx using ×40 objective of a Leica DMRA microscope. For both the survey of 238 molecules and the dose–response analyses, a minimum of three biological replicates were processed, and at least 20 worms per replicate were counted for the presence or absence of spheres and/or crystals.

To investigate the physicochemical property trends that are consistent with crystal-forming molecules and non-object forming molecules, we started with Chembridge's set of 500,000 "express-pick" molecules and applied the desired physicochemical filters. The selected range of crystal-like properties are: log Sw (approximately from −5 to −6); rotatable bonds (1 or 2); tPSA (≥50); H bond acceptors (≥4); H bond donors (0); mass (between 305 and 315). The selected range of no-object-like properties are: log Sw (approximately from −4.0 to −3.5); rotatable bonds (4–6); tPSA (≤40); H bond acceptors (1 or 2); H bond donors (1); mass (between 305 and 315). This left 122 molecules for the "crystal properties" list and 142 molecules for the "no-object" list. From these, we selected molecules with

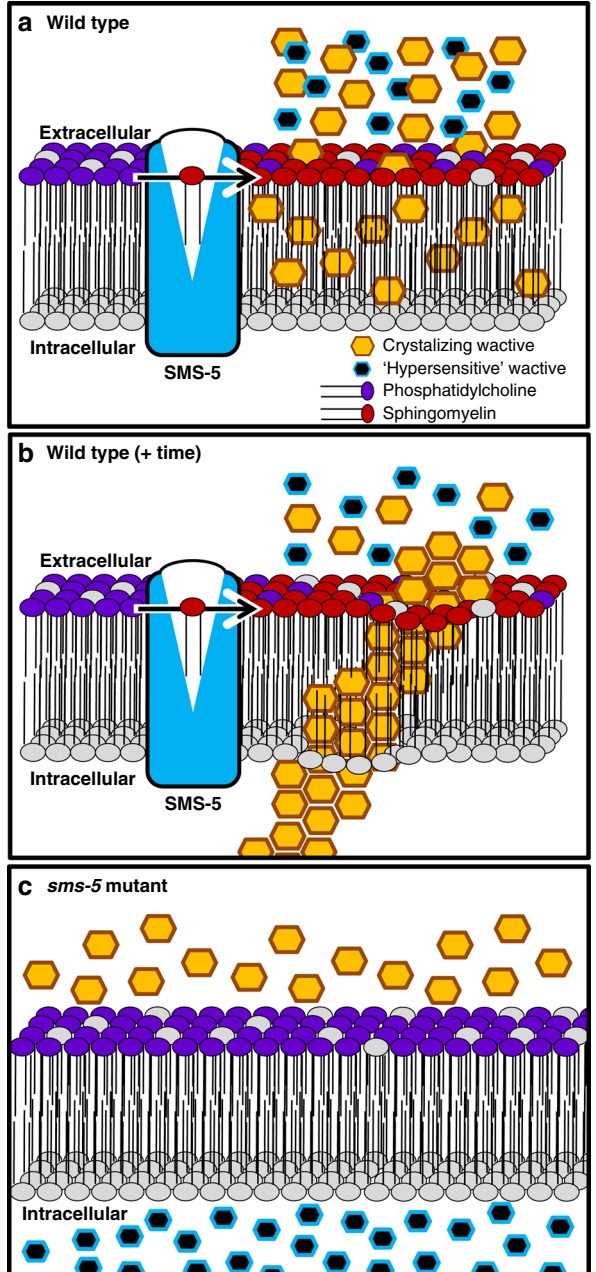

**Fig. 10** A model of SMS-5's role in small molecule accumulation. **a** SMS-5 is key in synthesizing sphingomyelin (SM) on the outer leaflet of the plasma membrane (PM) of the marginal cells of the anterior pharynx. The SM-rich membrane acts as a sink for relatively larger hydrophobic crystallizing wactives but a barrier to comparatively smaller molecules. **b** Over time, the hydrophobic wactives precipitate out of solution within the SM-rich PM of the wild-type marginal cells of the anterior pharynx. **c** The absence of SMS-5 results in less SM in the PM, which in turn reduces the accumulation of relatively large hydrophobic molecules. Because of decreased molecular density of a SM-poor PM, relatively smaller molecules can now penetrate the PM barrier

diverse properties within the limits of our search parameters. We conducted the tests in the same way as our initial survey of 238 molecules.

***C. elegans* viability assays**. Six-day viability assays were conducted by first generating a saturated culture of HB101. *E. coli* was concentrated twofold in liquid NGM and 40 µL of bacterial suspension was dispensed into each well of a flat-bottom 96-well plate. Compounds and dimethyl sulfoxide (DMSO) controls were pinned into

the 96-well plates using a 96-pin replicator with a 300-nL slot volume (V&P Scientific). The final concentration of DMSO in each well was 0.6% v/v. Approximately 20 synchronized L1 larval stage worms were added to each well of the 96-well plates in 10 µL of M9 buffer[36]. For some experiments, approximately 50 synchronized L1-stage worms were added to 80 µL of bacterial suspension in 20 µL of M9 buffer. Worms were synchronized to L1 stage through hypochlorite bleaching of gravid adults performed the previous day[36]. Assay plates were sealed with Parafilm and incubated for 6 days at 20 °C with shaking at 200 in a Newbrunswick scientific I26 incubator shaker. After 6 days, the plates were observed using a dissection microscope and the wells were categorized according to the number of viable adult and larval stage worms in each well. Wells with >50 animals after 6 days were categorized as over-grown. The number of worms in wells with approximately ≤50 worms were counted. All 6-day viability assays were completed in quadruplicate.

To analyze the ability of animals to recover from exposure to crystallizing wactives, synchronized L1s were incubated in 30 µM each wactive (or 1% DMSO control) using 50 L1s/well of a 24-well plate (Sarstedt) containing 100 µL of NGM plus HB101 bacteria ($t = 0$). The experimental samples were prepared with four technical replicates and the DMSO control with two technical replicates. At the indicated time points ($t = 0.5, 1, 2, 6, 24$, and 48 h), the entire liquid volume was withdrawn from the well and pipetted onto 6 cm MYOB plates seeded with OP50 and left to dry. The next day, the total number of L1s were counted on plate. Two days after that, the number of L4s and adults on each plate was recorded and expressed as a fraction of total number of L1s. Results are from two biological trials. All experiments were done at 20 °C.

To analyze the response of various strains to detergents, 25 synchronized L1 larvae were added to each well of a 24-well plate seeded with 25 µL OP50 and containing a final concentration of 0–1.6% detergent (Triton-X-100 or IGEPAL-630) dissolved in water (day 0). On day 6, the number of worms alive was counted and recorded. Results are from three independent trials with four technical replicates.

**Analysis of Evans Blue dye penetration into the pharynx**. L1-stage worms were incubated with 1% DMSO (control) or 60 µM wact-190 (in 1% DMSO) in liquid for 24 h. Worms were washed once with M9 and incubated with a final concentration of 0.1% Evans Blue dye in 500 µL of M9 in siliconized Eppendorf tubes for 4 h. Worms were then washed three times with M9 solution, suspended in 10 µL M9, and paralyzed for microscopy by adding 4 µL of 50 mM levamisole. Live worms were mounted on 3% agarose pad. Dye fluorescence was observed in TX2 channel of a Leica DMRA microscope at ×630 total magnification and quantified using the Fiji (ImageJ) software by calculating the mean gray value (MGV) in the anterior pharynx relative to the MGV in a nearby space in the frame devoid of worms. Experiments were done measuring at least six worms for each trial with a total of three biological trials.

**Molecular biology**. pPRHM1051 is the wrm0626dC03 fosmid, which contains the *sms-5* locus (W07E6.3), wherein the SMS-5 coding sequence is C-terminally tagged with YFP. This construct rescues mutant *sms-5*'s resistance to wact-190. pPRHM1051 was created using Oliver Hobert's fosmid tagging methodology[37]. Briefly, we PCR amplified the YFPint-FRT-galK-FRT cassette from the pBALU2 plasmid (a gift from Oliver Hobert) using the cassette-*sms-5* fusion primers, Fw (5'-cgagttccaaaaacgtgtcgacattgaaaaaatcacgaagatctttcgaaatgagtaaaggagaagaactttcac-3') and Rev (5'-gagattttttattcaattttttgttagcaaaaataaattgttcagcctaatTTAtttgta-tagttcatccatgccatg-3'). We transformed the bacterial strain carrying the wrm0626dC03 fosmid (a gift from Don Moerman) with the amplified PCR product and followed the detailed recombineering protocol described in Tursun et al. (2009) to integrate the YFP tag directly into the *sms-5*-containing fosmid[37].

pPRHZ1138 (pgp-14p::SMS-5B(genomic)::FLAG::mCherry) (aka SMS-5(MC)) construct is based upon pPRHM1065 (myo-2p::SMS-5::FLAG::mCherry). Briefly, we used primers HZ4pgp14hind.for (5'- aaattaagcttcaacagagagcaaggtg-3') and HZ3pgp14prokpn.rev (5'-aaattggtaccgtttaattatcgtacatcg-3') to amplify by PCR a 1.6-kb *pgp-14* promoter fragment from the pPRZH1160 (the *pgp-14* fosmid (WRM065dH09) 7.748 kb NaeI/SacI fragment in pKS) construct. We then cut the 1.6-kb PCR fragment with HindIII and KpnI. We cut out the *myo-2* promoter in pPRHM1065 by using HindIII and KpnI and purifying the 7.5-kb fragment and ligating it to the 1.6-kb *pgp-14* promoter. We verified the construct through restriction digest analyses.

**Sample preparation for mass spectrometric (MS) analyses**. We grew a mixed stage population of wild-type (N2) *C. elegans* and collected embryos through hypochlorite bleaching of gravid adults performed the previous day[36]. Embryos were allowed to hatch overnight in M9 buffer. The resulting synchronized L1s were harvested the next morning. L1s were then grown on 10 cm plates, with 10,000 L1s per plate, and incubated at room temperature (RT) for 48 h. Worms were then washed off the two plates with M9 buffer, collected into a 15-mL conical tube, and washed three times. In parallel to this, we prepared a "drug"-incubation buffer by first inoculating 400 mL of LB with 50 µL of an overnight culture of *E. coli* (HB101), grew it overnight, and centrifuged 50 mL of this fresh culture for 10 min at 2100 × g. We decanted the LB and rinsed the bacteria in ~50 mL of NGM buffer once and then resuspended the bacteria in 25 mL of NGM. We resuspended 20,000

worms in 1 mL of the NGM+HB101 solution in either 1% DMSO or 30 μM of small molecule (to a final DMSO concentration of 1%) in 1.5-mL siliconized microcentrifuge tubes. We then incubated the tubes for 4 h at 20 °C on a nutator. Thereafter, we washed the worms five times with ice-cold M9 buffer, keeping the samples on ice as much as possible. We removed the M9 and flash froze the samples using liquid nitrogen and stored the worms at −80 °C until the samples were ready to process by MS.

To perform an methyl tert-butyl ether (MTBE) extraction of small molecules from the worm lysate, we first lyophilized the worm pellet and resuspended it in 200 μL of 0.1 M NaCl, followed by sonication for 5 min at 100 W in a Misonix cup sonicator. Samples were then spiked with 43 ng of internal standard (triamcinolone acetonide) and extracted twice with MTBE (sample: MTBE, 1:5, v/v). Organic layers were pooled and evaporated to dryness under $N_2$. Extracts were re-suspended in 200 μL of high-performance liquid chromatography (HPLC)-grade methanol and analyzed as described below.

**MS analyses of wactive accumulation**. The samples were analyzed by LC/MS/MS using a 6410 LC/MS/MS instrument (Agilent Technologies) with an ESI source in positive ion mode. Samples were separated on a Zorbax XDB-C18 column (4.6 × 50 mm, 3.5 μm) at 0.4 mL/min. The mobile phase consisted of HPLC-grade water (A) and methanol (B) both containing 5 mM NH₄Ac. The following gradient was run: 0–1 min, 60% (B); 1–3.3 min, 60–100% (B); 3.3–7 min, 100% (B); stop time, 10 min; post-time, 5.5 min. MS parameters were as follows: nebulizer pressure 35 psi, drying gas (nitrogen) 10 L/min, VCap 6000 V, Delta EMV 800 V, column temperature 40 °C, and drying gas temperature 350 °C for all compounds. For each of the following molecules, the following transitions were measured using multiple reaction monitoring (following the molecule's name, the run-time (min), MRM, Fragmentor (voltage), and Collision Energy (voltage) are provided): triamcinolone acetonide, 5.7, 435 → 415, 108, 5; wact-190, 7.7, 330 → 107, 155, 22; wact-171, 7.3, 386 → 131, 45, 12; wact-498, 7.4, 346 → 238, 45, 40; wact-406, 5.9, 240 → 119, 45, 16; wact-455, 6.2, 254 → 105.1, 45, 40; wact-519, 6.8, 324 → 240.2, 45, 16.

**GFP-NT-Lysenin protein production and testing**. GFP-NT-Lysenin protein[38] was produced by first transforming BL21 (DE3) Rosetta cells with pMAL-C2-GFP-NT-Lysenin vector (a gift from Dr. Gregory Fairn, University of Toronto) and cultured in 1 L of LB broth at 37 °C until OD₆₀₀ reached 0.6[39]. Expression was induced with 1 mM IPTG for 6 h at 25 °C. Cells were frozen overnight at −80 °C. Cells were lysed using B-PER reagent (Thermo Fisher Scientific) according to the manufacturer's instructions in the presence of excess MgCl₂, DNase I, lysozyme, and protease inhibitor cocktail. Crude cell lysate was passed through a Sepharose column containing 5 mL amylose resin (New England Biolabs) and the MBP-tagged GFP-NT-Lysenin fusion protein was eluted with fresh 10 mM maltose in 1× phosphate-buffer saline (PBS; pH 7.4). All purification steps were carried out at 4 °C. Sodium dodecyl sulfate-polyacrylamide gel electrophoresis confirmed the presence of the 90 kDa fusion protein, which was stored at 4 °C until use.

To test the efficacy of the GFP-NT-Lysenin protein, we created an artificial lipid bilayer of 1,2-dioleoyl-sn-glycero-3-phosphocholine either with or without brain SM (1:1) prepared in 10 mM HEPES buffer and 150 mM NaCl, pH 7.4. DiI-C₁₈ (Invitrogen, Oakville, ON) was added to the lipid mixtures from a stock solution to a final concentration of 1 mol%[40]. To create the bilayers, the lipid mixture was dried by rotary evaporation for 1.5 h. The resulting film was rehydrated in the HEPES buffer to a concentration of 1 mM and sonicated at 65 °C for 20 min to produce small unilamellar vesicles (SUVs). The lipid bilayer was formed by vesicle fusion onto freshly cleaved V1 grade muscovite mica affixed using UV-curable adhesive (Norland Optical Adhesive 63, Norland Products, Cranbury, NJ) to a glass-bottom Willco dish. The fluid cell was incubated for 5 min at 50 °C with 400 μL HEPES buffer and 5 mM CaCl₂ to make the surface more hydrophilic prior to adding 100 μL of the SUV suspension. To facilitate liposome fusion and bilayer formation, the freshly cleaved mica sealed in the fluid cell was incubated for ~10 min with HEPES buffer (10 mM HEPES, 150 mM NaCl, pH 7.4, 1 M CaCl₂) prior to the introduction of ~500 μL of hydrated liposomes heated to ~70 °C. After ~30 min of incubation at RT to allow for bilayer formation, the fluid cell was flushed with liposome-free buffer. If an excessive number of vesicles were observed under TIRF illumination, 1 mL aliquots of buffer were exchanged from the dish as needed. If a continuous bilayer was not present, 100 μL aliquots of 1 mM lipid stock were exchanged for 100 μL buffer from the dish as needed. A home-built polarized TIRF microscope built around an Olympus IX70 inverted microscope that accommodates multiple excitation laser lines was utilized for pTIRF microscopy. The bottom of the dish was brought into focus under ambient lighting under oil immersion using a ×60 1.45 NA TIRF objective. Appropriate filters were then inserted and a region of the supported bilayer was brought into focus under pTIRF illumination. Images were captured with an Evolve 512 EMCCD camera (Photometrics, Tucson, AZ) controlled by Micro-Manager (http://micro-manager.org). Fluorescent probes were excited by parallel (s) or perpendicular (p) polarized light, relative to the substrate surface, through the rotation of a half-wave plate in the excitation path. Images were acquired before and after the addition of 20 μg/mL GFP-NT-Lysenin protein to the imaging chamber.

**Staining worms with GFP-NT-Lysenin protein**. Synchronized adults were fixed in 1.5-mL Eppendorf tubes using the Modified Finney-Ruvkun protocol[41]. Briefly, worms were washed with chilled M9 buffer three times, removing as much bacteria as possible. The worm pellet was incubated in ice for 30 min and then mixed with 2× MRWB (5% methanol) and freshly prepared paraformaldehyde (PFA; final concentration—4%), freeze–thawed in an ice-ethanol bath four times, and then incubated on ice for 2 h. The worm suspension was then reduced and oxidized with 1% beta-mercaptoethanol and 10 mM dithiothreitol and 0.3% hydrogen peroxide, respectively. The following changes were made to the protocol: All washes were done with tris-triton buffer (TTB) lacking triton-X-100; triton-X-100 was also replaced with 1× PBS in Buffers A and B in the final steps. The quality of fixation was tested by actin staining of fixed worms following 2 h incubation at RT with 0.328 nM phalloidin.

To stain worms with GFP-NT-Lysenin, 20 μL of fixed worms were suspended in 0.5 mg/mL of GFP-NT-Lysenin in the final volume of 100 μL of 1× PBS in a 0.6-mL tube. Fixed worms were incubated at 4 °C for 4 h in the dark with constant mixing. Worms were washed two times with 5.5 mL ice-cold 1× PBS. Ten microliters of worms were mounted on a glass slide containing 3% agarose and a cover-slip was applied. Animals were imaged in the GFP channel using ×20 magnification and identical exposure times for all strains. The experiment was repeated at least three times. Fluorescence intensity in the anterior pharynx was quantified using the ImageJ software. For each worm, fluorescence intensity was determined by subtracting the background MGV from MGV of the procorpus. Student's T test was performed on mean values of the different trials using GraphPad 6.0.

**Cholesterol-limitation assay**. Worms were grown on MYOB plates seeded with E. coli (OP50). Gravid adults were bleached and the resulting embryos were incubated overnight at 20 °C to yield synchronized L1 parents (P0s). The next day, ~80 synchronized L1 P0s were plated onto modified NGM plates (35 mm) in triplicate. The modified NGM recipe is used to deplete its contents of sterols, and is described in ref. [10]. Briefly, the agar is replaced with agarose (Froggarose) and the peptone is extracted with ether three times and dried in a fume hood overnight. The plates incorporate either standard (5000 ng/mL, "+xol") or low (50 ng/mL, "−xol") concentrations of cholesterol. We did not prepare plates without choles-terol because resulting F1s arrest as L1s, regardless of the genotype we tested. All plates are then seeded with 50 μL of an overnight OP50 LB culture.

Three days after seeding plates with L1 P0s, 20 gravid adults are transferred to fresh plates in triplicate and allowed to lay eggs. P0s raised on +xol plates are transferred to new +xol plates; P0s raised on −xol plates are transferred to new −xol plates. After 6 h of egg-laying, the P0s are removed. We define this as "day 0".

On day 4, the number of F1 L4s and/or adults on the plates are counted. The experiment was repeated at least three independent times (each with three technical replicates) at 20 °C. For each strain, we measure the impact of xol limitation by calculating the ratio of F1 L4s/adults on the −xol plates relative to the F1 L4s/adults on the +xol plates (to account for fecundity/growth issues unrelated to xol). Note that we store our filter-sterilized cholesterol stock (in 100% ethanol) in a 50-mL falcon tube sealed with parafilm at 4 °C to prevent evaporation.

**NBD-cholesterol accumulation assay**. NBD-cholesterol accumulation assays were performed by modifying a previously published protocol[28] as follows. An overnight OP50 culture was pelleted and diluted with autoclaved water to which was added 22-NBD-cholesterol dissolved in ethanol (Thermofisher Inc.) at a concentration of 200 μM and vortexed vigorously. One hundred and seventy-five microliters of the OP50 + 22-NBD-cholesterol mixture was added to 6 cm NGM Petri plates containing 5 μg/mL (~13 μM) cholesterol made on the same day and then dried in a laminar flow hood for 1 h. The next day (defined as "day 0"), 25 synchronized L1-stage larvae were plated onto each of three 6 cm NGM Petri plates for each strain. On day 6, worms were removed from plates and washed three times with M9. After the final wash, 200 μL of worm pellet was added to an Eppendorf tube, to which was added 75 μL of 20% of freshly prepared PFA diluted to a final volume of 500 μL with M9 (to final concentration of 3%). Worms were fixed for 30 min at RT. Next, the worm pellet was washed twice with 1 mL of a 100 mM glycine solution to quench the PFA. A minimal volume of concentrated worm pellet (~5 μL) was then deposited onto a 3% agarose pad on a glass slide. A small drop of a fluorescent mounting medium (Thermo Scientific™ Richard-Allan Sci-entific™ Cytoseal 280) was added to the sample before adding a coverslip. For each sample, 30–70 worms were imaged using identical exposure times and ×10–×20 objectives. ImageJ was used to quantify fluorescence intensity in the anterior and posterior halves of the worm gut, along with background areas next to the anterior and posterior halves to control for differences in background signals. The fluor-escence intensity is expressed as MGV. The experiment was performed at least three times. Except where noted, worms were incubated at 20 °C.

**Emission spectra of wactives**. Emission spectra for select wactives were measured by making a 50-μL solution of each compound at a concentration of 250 μM in double-distilled water. A fluorescence intensity scan was performed in 96-well plate format using an Infinite M200 Pro microplate reader (Tecan Life Sciences) with an excitation wavelength set at 390 nm and the range of emission wavelength mea-sured at 425–700 nm with 25-nm steps. The resulting intensity values were nor-malized against a solvent blank.

**Transmission electron microscopic (TEM) analyses**. Approximately 1000 synchronized L1-stage worms were added to 60-mm plates containing 60 μM of wact-190 or 1% DMSO vehicle control seeded with OP50 bacteria. Twenty-four hours later, worms were collected and prepared for TEM[42]. Live animals were loaded into a metal planchette in a slurry of *E. coli* and fast frozen under high pressure using a Balt-tec HM 010 freezing device, placed into 2% osmium tetroxide, 0.1% UAc, 2% dH$_2$O in acetone at −90 °C for 96 h using a Boeckler freeze substitution unit, slowly warmed (5 °C/h) to −30 °C, held 16 h, then slowly warmed (5 °C/h) to 0 °C, then rinsed in cold acetone at 0 °C, and again rinsed several more times at RT before infiltration into plastic resin over 3 days (HardPlus Embed812). Once fully infiltrated, the samples were cured at 60 °C for 2 days and thin sectioned for views at high resolution using a Philips CM10 electron microscope. Some samples were viewed in cross-section, and others in lengthwise section for comparison. Control animals were exposed to DMSO without drug for 24 h.

**Statistics and graphs**. Except where indicated, statistical differences were measured using a two-tailed Student's *T* test. Violin plots were generated using the online tool BoxPlotR (http://shiny.chemgrid.org/boxplotr/).

**Animal ethics statement**. We (the authors) affirm that we have complied with all relevant ethical regulations for animal testing and research. Given that our experiments focused exclusively on the invertebrate nematode worm *C. elegans*, no ethical approval was required for any of the presented work.

## Data availability

Any primary data generated or analyzed during this study that are not included in this published article and its supplementary information files are available upon request (i.e., whole-genome sequence, etc).

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

## Acknowledgements

We thank Greg Fairn and Masashi Maekawa for reagents and guidance in characterizing sphingomyelin abundance and localization in the worm; Hong Zheng for building the p1138 construct and Kevin Chan for microinjection and integration work; Don Moerman and Oliver Hobert for fosmids and fosmid recombineering tools; David Hall, Zeynep Altun, and Chris Crocker for permission to modify Worm Atlas schematics[43]; Andrew Burns for helpful comments on the work and mining Chembridge libraries; and Andy Fraser and Michael Schertzberg for whole-genome sequence analysis; Lindy Holden-Dye and Fernando Calahorro Nunez for preliminary analyses; and the *C. elegans* Genetics Centre and Shohei Mitani for mutant strains. This work was supported by CIHR grants (376634 and 313296) and a CRC to P.J.R., an NSERC (RGPIN 03666-14) to C.L.C., and an NIH grant (OD 010943) to D.H.H.

## Author contributions

M.K. discovered crystal and sphere formation and generated and analyzed all primary data herein that is not included in the contributions by other authors listed below. H.M. performed the initial screens and characterization of wact-190-resistant mutants. L.M. performed the mass spectrometric analyses. D.H. performed the GFP-NT-Lysenin stains and analyses. K.C.Q.N. performed the TEM analyses. M.Y., J.K., and R.B. conducted detailed viability analyses on the wact-190-resistant mutants in the background of wact-190 and the entire wactive library. A.M.W. conducted the artificial lipid bilayer analysis of GFP-NT-Lysenin staining. K.S. conducted some crystal formation counts. C.M.Y. supervised and secured funding for the artificial bilayer work. C.L.C. supervised and secured funding for the mass spectrometric work. D.H.H. supervised and secured funding for the TEM work. P.J.R. supervised and secured funding for the project, and together with M.K., wrote the manuscript and composed the figures.

## Additional information

**Competing interests:** The authors declare no competing interests.

