## [Peer Review File · Nature Communications]

Reviewers' comments:

Reviewer #1 (Remarks to the Author):

This manuscript presents very novel and intriguing data on the important model organism *C. elegans*. The data pertain both to the use of this organism for drug discovery and with regard to the basic biology of nutrient acquisition. I admire the authors for presenting this work, but have a number of major concerns about the interpretation and presentation of these data.

1. The physical-chemical parameters summarized in Figure 1 D overlap extensively for all three classes of compounds; I see no solid evidence that these parameters can be used to predict whether an unknown will form aggregates or understand the basis for aggregate formation for the current set of wactives. I believe this aspect of the manuscript needs to be greatly toned down.

2. Page 6: Please clarify if the spheres and crystals are intracellular or extracellular at the start of formation. .

3. Page 6: The authors mention "day to day variability" in response to wactives but provide no quantitation or explanation for its basis. More details are necessary.

4. Is the lethal effect reversed if the worms are removed from the presence of the wactive? Is the ability to recover from exposure time-dependent?

5. Do the physical-chemical characteristics of cholesterol fall within the wactive range? Would a high concentration of cholesterol in the medium be predicted to accumulate in aggregates? Would it be lethal? If the formation and toxicity of at least some wactives are linked based on their chemical properties, and if similar biology is thought to underlie the accumulation of similar molecules, this should be tested.

6. I do not think it proper to refer to pgg mutants and then defer their description to another manuscript. If they are relevant for this phenotype, they should be included here.

7. Page 12: The concentration-response curves shown in Figure 7d prove that, for a subset of wactives, the toxicity is independent of particle formation: the shape of the concentration-response curve is the same whether or not crystals form. Why do the authors propose that these molecules have two mechanisms of action, one based on crystal formation and one independent of that?

8. Page 15: bacteria in the environment can incorporate cholesterol into their membranes, and this nutrient would almost certainly be otherwise associated with particles in the environment. What is the evidence that "free" cholesterol (fully dissolved) is available to this organism in its ingesta? This is not compatible with the hypothesis of the function of marginal cells.

9. Page 16: "blockage of the alimentary tract": this concept was not introduced in the manuscript; either expand or delete.

10. Minor concerns: "precious" is not the word I would use; "essential scarce" works better for me. Please use concentration instead of dose in the context of these experiments. Page 8: "emanating" is not the word I would choose; perhaps "breaching"? Page 11: "at least 169..."; were there more? Second to the last line on page 11: fewer instead of less.

Reviewer #2 (Remarks to the Author):

This manuscript investigates a phenomenon that many (including myself) might have dismissed as uninteresting and trivial – the formation of crystals in the pharynx of *C. elegans* treated with hydrophobic small molecules – in order to uncover a novel mechanism by which worms filter important small molecules from their environment, such as cholesterol. This work has important implications for small molecule screening in *C. elegans* because it shows that many of the compounds that may be identified in screens as being lethal to worms might work through a common mechanism of crystal formation, and in order to find compounds that work through other mechanisms, one can screen in an *sms-5* background, which largely prevents crystal formation.

I found the conclusions of this paper to be well-supported, and so I only have a few comments.

It would be useful for the readership to know how common these crystal forming compounds are. The authors found that about 30% of the 238 wactives in Fig. 1 gave crystals or spheres, but perhaps these high percentages have something to do with the particular choice of the 238 wactives that were screened. In particular, it appears that the percentage of the 238 wactives in Fig. 1 that kill the worms is higher than the percentage of the 508 wactives in Fig. 6. Perhaps in Fig. 1, it would be more relevant to talk about the percentage of the wactives that kill worms that also give rise to crystals or spheres?

For the sms-5::flag-mcherry integrated reporter, it is referred to several names, including aMCp::sms-5::rfp and sms-5::flag-mcherry and sms-5(MC). I think these are all the same? It would be less confusing if one name were used throughout.

The authors should show that the sms-5::flag-mcherry integrated reporter is specifically expressed in the anterior marginal cells. The image in SI Fig. 8 is too low resolution to tell.

Page 28 – For the molecular biology section, the authors say that detailed construct plans are available on request. However, the authors do not have that many constructs and so should easily be able to include these plans in the methods.

For SI Fig. 2, the arrows should be green to be consistent with other figures.

For SI Fig. 3, how do the authors know that they are imaging the marginal cells as opposed to the intervening muscle cells? In figure A", is the tissue above and below the central lumen supposed to be marginal cells? It would be useful to have a figure to help orient the reader.

In SI Fig. 5 B', C', D', and E', the images should be rotated clockwise 90 degrees to make the orientation more consistent with the images above.

In SI Fig. 6, the labeling could be made more clear by including bars above and to the side of the images with grouped labels for the different types of images. The results described within the images should be placed just below the images to make the text more legible.

Reviewer #3 (Remarks to the Author):

The authors describe the accumulation of some worm active substances (wactives) in the marginal cells of the pharynx. It is concluded that these are more hydrophobic substances from the pool. A genetic screen identified the sphingomyelin pathway to be involved in the resistance/accumulation to wactive-substances. In particular, SMS-5 (sphingomyelin syntase-5) is involved in wactive accumulation. The authors also show that SMS-5 might be responsible for cholesterol uptake.

The experiments presented are performed thoroughly with all the controls required. The whole study, however, is purely descriptive, giving no clue how sphingomyelin can be involved in the accumulation of wactives. Even the association of cholesterol uptake with SMS-5 activity does not provide any insight into the molecular mechanism of the process. In addition, there is whole a zoology of wactive molecule classification into crystalizing and non-crystalizing ones.

Reviewer #1 (Remarks to the Author):

Reviewer 1-Opening Remarks: This manuscript presents very novel and intriguing data on the important model organism *C. elegans*. The data pertain both to the use of this organism for drug discovery and with regard to the basic biology of nutrient acquisition. I admire the authors for presenting this work, but have a number of major concerns about the interpretation and presentation of these data.

Author's Response: We thank the reviewer for their kind words and encouragement.

Reviewer 1-Comment 1: The physical-chemical parameters summarized in Figure 1 D overlap extensively for all three classes of compounds; I see no solid evidence that these parameters can be used to predict whether an unknown will form aggregates or understand the basis for aggregate formation for the current set of wactives. I believe this aspect of the manuscript needs to be greatly toned down.

Author's Response: We have modified the relevant paragraph to emphasize that we are looking at trends and not constructing a predictive model. The revised text is pasted here (changes in bold):

*We investigated whether there are basic physicochemical features of the wactives that **correlate** with crystallization and sphere formation. We found that both crystal-forming and sphere-forming wactives are less hydrophilic (and have fewer hydrogen bond donors, which is related to hydrophilicity) **on average** than the wactives that do not generate unusual objects ($p \leq 0.01$) (Fig 1d). This **trend** is consistent with the molecules precipitating out of solution when concentrated within the animal. One **trend** that **may** distinguish the crystal-forming compounds from those that form spheres is that the former have a greater topological polar surface area **on average** ($p < 0.05$) (Fig 1d)....*

To probe these trends further, we tested the accumulation of a set of 10 newly purchased molecules whose physicochemical properties fall on one side of the trend towards crystallization and another set of 10 newly purchased molecules whose physicochemical properties fall on the other side of the trend (away from object formation). The molecules were purchased from Chembridge Inc and had the following properties:

crystal-like properties

Log Sw (~-5 to -6)
Rotatable bonds (1 or 2)
tPSA (50 or more)
H bond acceptors (4 or more)
H bond donors (0)
Mass (between 305 and 315)

non-object properties

Log Sw (~-4.0 to -3.5)
Rotatable bonds (4-6)
tPSA (40 or less)
H bond acceptors (1 or 2)
H bond donors (1)
Mass (between 305 and 315)

We have summarized this new experiment in the final paragraph in the first results as follows:

To probe these trends further, we tested a set of 10 newly purchased molecules whose physicochemical properties fall on one side of the trend towards crystallization and another set of 10 newly purchased molecules whose physicochemical properties fall on the other side of the trend (away from object formation). Of the 10 molecules with crystal-like physicochemical properties, two crystalized in the anterior pharynx. By contrast, none of the molecules with non-object like physicochemical properties formed objects in the pharynx. That 20% of the molecules with crystal-like physicochemical properties form crystals is an enrichment of 147-fold over what is expected by random chance (see Supplemental Dataset 2 for details).

Reviewer 1-Comment 2: Page 6: Please clarify if the spheres and crystals are intracellular or extracellular at the start of formation.

Author's Response: In the original submission, we present three lines of evidence that suggest that objects form intracellularly. First, animals that hatch *in utero* (in mothers that have been bathed in the compound), and have never been exposed to the outside of the worm, form crystals. This is strongly supportive of the idea that objects form inside of the hatchling (i.e. intracellularly) and are not seeded by obvious objects from the worm's environment.

Second, a close inspection of the images at the 30 minute time point of our wact-190 time course (Figure 3B) show a close association with what looks like the basal lamina or the plasma membrane on the cellular side of the marginal cells (as opposed to the luminal side of the marginal cells). We see no birefringent objects floating around the lumen in these figures or in the images of the hatchlings *in utero* (Supplemental Fig 2).

Third, the TEM images at the 24 hour time point (Fig 3D and Supplemental Fig 3A) clearly show the objects in the cytoplasm of the marginal cells and not in the extracellular milieu.

In response to the reviewer's query, we tested the hypothesis that external crystal formation is sufficient to seed crystal formation in the anterior pharynx (or anywhere else). We examined 17 small molecules that precipitate in our liquid assay media but fail to perturb the development or viability of the worms. If external crystal formation is sufficient to seed crystal formation in the pharynx, then each of these 17 molecules should result in crystals in the pharynx. Inspection revealed that none of these molecules form crystals or spheres in the pharynx. This indicates that external crystal formation is insufficient to seed crystal formation in the pharynx.

Together, our original and new results are all consistent with the idea that the objects form in association with the cells and are not seeded from extracellular objects. We have revised the relevant section and added a new supplemental figure (now Supplemental Figure 4) in the updated manuscript as follows (changes in bold):

Three observations argue against this. First, the birefringent crystalline objects that can be seen early in the time course experiment are not found in the lumen of the channels, but appear to be associated with the cytoplasmic face of the basal lamina of the marginal cells and possibly its plasma membrane (see the 30 minute time point in Figure 3). Second, we found that crystals accumulate in the pharynx of young larvae that hatch within the parent's uterus and are not exposed to the external media (Supplemental Fig 3). Third, we tested whether small molecules that precipitated in the media but fail to perturb the worm's development lead to crystal formation in anterior pharynx and found that none of the 17 molecules of this class formed objects in the pharynx (Supplemental Fig 4). Together, these observations argue against the idea that crystals are seeded from crystals in the media. Instead, it is more likely that soluble small molecule accumulates in the tissue of the anterior pharynx, precipitates out of solution and forms crystals or spheres therein.

Reviewer 1-Comment 3: Page 6: The authors mention "day to day variability" in response to wactives but provide no quantitation or explanation for its basis. More details are necessary.

Author's Response: We realize that this note adds irrelevant confusion to the section, so we removed the sentence from the revised text.

Reviewer 1-Comment 4: Is the lethal effect reversed if the worms are removed from the presence of the wactive? Is the ability to recover from exposure time-dependent?

Author's Response: We investigated the reviewer's query using three crystalizing wactives (wact-190, wact-416, and wact-498) at 30 μ M (plus a DMSO-only control) and found that indeed, the lethal effect of the wactives can be prevented for most of the animals if they are removed from the wactive before the 48 hour time point; thereafter the vast majority of animals do not recover from exposure to the wactive.

We added the following text to the results section:

'We also asked whether the lethality associated with crystal-forming wactives can be avoided if the animals are removed from the wactive. We found that the lethal effect of the wactives can be prevented for most of the animals if they are removed from the wactive during the first two days of co-incubation with the molecule; thereafter, most animals do not recover from exposure to the wactive (Supplemental Fig 2).'

We updated the methods section to describe this experiment and have added a new Supplemental Fig 2 (and have consequently changed the subsequent figure numbers):

Reviewer 1-Comment 5a: Do the physical-chemical characteristics of cholesterol fall within the wactive range?

Author's Response: We have plotted the physicochemical properties of cholesterol relative to the crystal forming compounds below. Except for the number of hydrogen bond donors, it is apparent that cholesterol's properties are on the tails of the range of the wactive.

Of note, cholesterol's topological polar surface area (tPSA) is relatively low, and its number of rotatable bonds is relatively high, compared to the crystalizing wactives. In the manuscript, we suggest that a large tPSA and a relatively stiff (i.e. lower rotatable bond number) structure may be two features that allow wactives to crystalize. Hence, for these reasons, and the fact that there are cholesterol distribution mechanisms in animals (see below), we would be surprised to see cholesterol crystalize out of solution in the marginal cells (or anywhere else in the worm).

As a reminder, in the original and revised manuscript, our stated rationale for investigating whether SMS-5 (+) marginal cells play a role in cholesterol absorption was simply that it was a hydrophobic small molecule nutrient.

Figure Note: The properties of cholesterol are indicated with a red asterisks.

Reviewer 1-Comment 5b: Would a high concentration of cholesterol in the medium be predicted to accumulate in aggregates? Would it be lethal? If the formation and toxicity of at least some wactives are linked based on their chemical properties, and if similar biology is thought to underlie the accumulation of similar molecules, this should be tested.

Author's Response: *C. elegans* (and other animals) has evolved cholesterol transport systems (NCR-1, NCR-2, etc) to distribute cholesterol both intracellularly and throughout the body. Hence, the worm is capable of recognizing the molecular structure of cholesterol and distributing a range of concentrations of the molecule. This, together with cholesterol's physicochemical properties that deviate from crystalizing wactives, suggested to us that it may not crystalize in the marginal cells.

We tested whether increasing the concentration of cholesterol up to its limit of solubility in standard (NGM) *C. elegans* growth plates can elicit the formation of unusual objects in wild type L1 and L4 animals. We found that cholesterol did not visibly accumulate in the pharynx or anywhere else. We have included an example of pharynxes from animals incubated in standard amounts of cholesterol (XOL) and in 8 fold higher XOL concentrations (cholesterol's limit of solubility in NGM media) here, but have not included these data in the revised manuscript.

Note on the Figure above: Depending on the orientation of the animal, the regular birefringence of the pharynx musculature becomes apparent. This is clearly seen in the upper right panel and is easily distinguished from much more intense and irregular nature of small molecule crystal formation.

Reviewer 1-Comment 6: I do not think it proper to refer to *pgp* mutants and then defer their description to another manuscript. If they are relevant for this phenotype, they should be included here.

Author's Response: We appreciate the reviewer's curiosity and share their enthusiasm about the *pgp-14* mutants. However, we have three reasons not to include additional details about *pgp-14*'s chemical-genetic interactions in this manuscript: i) We are actively working on better understanding the mechanism by which *pgp-14* contributes to the observed phenotypes and want to better flesh that out before submitting the work for publication; ii) The current (SMS-5-centric) manuscript is already large (9 figs; 10 sup figs; 3 supplementary datasets) and including more sections and figures will make it unwieldy; and finally, iii) It is not unusual to present loci that are not followed up in detail in papers that involve a forward genetic screen. There are many *C. elegans* papers in which a forward genetic screen was done and only a subset, or even just one, of the resulting mutant genes were discussed in detail. Here are several examples (from our lab and others):

In this paper, we performed a genetic screen, described many loci, but left the detailed description of *madd-2* for its own respective paper:

The genetic screen: Alexander, M.*, Chan, K.K.*, Byrne, A.*, Selman, G., Lee, T., Ono, J., Wong, E., Puckrin, R., Dixon, S.J., and Roy, P.J.§. 2009. An UNC-40 Pathway that Directs Postsynaptic Membrane Extension in *Caenorhabditis elegans*. *Development*, 136, 911-922. PMID: 19211675

The characterization of *madd-2*: Alexander, M., Selman, G., Seetharaman, A., Chan, K.K., D'Souza, S.A., Byrne, A.B., and Roy, P.J.§. 2010. MADD-2, a Homologue of the Opitz Syndrome Protein MID1, Regulates Guidance to the Midline through UNC-40 in *Caenorhabditis elegans*. *Developmental Cell*, 18, p961-972. (DOI 10.1016/j.devcel.2010.05.016). PMID: 20627078

In this paper, we performed a genetic screen, described many loci, but left the detailed description of *madd-3* for its own respective paper:

The genetic screen: Seetharaman, A., Selman, G., Puckrin, R., Barbier, L., Wong, E., D'Souza, S.A., and Roy, P.J.§. 2011. MADD-4 is a Secreted Cue Required for Midline-Oriented Guidance in *Caenorhabditis elegans*. *Developmental Cell* 21, p669-680. PMID:22014523

The characterization of madd-3: D'Souza, S.A., Rajendran, L., Bagg, R., Barbier, L., and Roy, P.J.§. 2016. The MADD-3 LAMMER Kinase Interacts with a p38 MAP Kinase Pathway to Regulate the Display of the EVA-1 Guidance Receptor in *Caenorhabditis elegans*. *PLoS Genetics*. DOI:10.1371/journal.pgen.1006010. PMID: 27123983

Other randomly chosen examples:

The characterization of one mutant in detail out of six isolated (see page 4, third paragraph):

Oikonomou, G., Perens, E. A., Lu, Y., Watanabe, S., Jorgensen, E. M., and Shaham, S. (2011). Opposing Activities of LIT-1/NLK and DAF-6/Patched-Related Direct Sensory Compartment Morphogenesis in *C. elegans* (Supplemental Data). *PLoS Biol.* 9: e1001121.

Just one mutant gene from a screen described: Fujisawa K, Wrana JL, Culotti JG. 2007. The slit receptor EVA-1 coactivates a SAX-3/Robo mediated guidance signal in *C. elegans*. *Science*. 317(5846):1934-8.

Pursued just two mutant genes from a collection of 84 mutants isolated in a screen: Kim K, Sato K, Shibuya M, Zeiger DM, Butcher RA, Ragains JR, Clardy J, Touhara K, Sengupta P. 2009. Two chemoreceptors mediate developmental effects of dauer pheromone in *C. elegans*. *Science*. Nov 13;326(5955):994-8.

Pursued just one mutant gene from a collection of 5 mutants isolated in a screen: Wu Q, Cao X2, Yan D2, Wang D3, Aballay A. Genetic Screen Reveals Link between the Maternal Effect Sterile Gene *mes-1* and *Pseudomonas aeruginosa*-induced Neurodegeneration in *Caenorhabditis elegans*. *J Biol Chem*. 2015 Dec 4;290(49):29231-9.

Reviewer 1-Comment 7a: Page 12: The concentration-response curves shown in Figure 7d (*sic*) prove that, for a subset of wactives, the toxicity is independent of particle formation: the shape of the concentration-response curve is the same whether or not crystals form.

Author's Response: We agree with the reviewer and explicitly call attention to this fact in the original and revised manuscript by writing, '*In all cases, we found that the *sms-5* mutant resists crystal formation, but half of these wactives are still able to kill the mutant (Fig 7b).*'

Reviewer 1-Comment 7b: Why do the authors propose that these molecules have two mechanisms of action, one based on crystal formation and one independent of that?

Author's Response: Let us paraphrase the reviewer's question so that it is clear that we are addressing the correct issue: For the group of compounds that kill and crystalize wild type animals but only kill (and not crystalize in) *sms-5* mutant animals, why are the authors suggesting that: a) this group of molecules may have two mechanisms of killing (one that is crystal-dependent, and one that is crystal-independent), instead of: b) a single mechanism of action that is independent of crystal formation (and therefore crystal formation of this group is simply incidental)?

The reason why we suggest two mechanisms of action for this group of molecules is because we have provided four lines of evidence that crystals are capable of killing *C. elegans*. We remind the reader of this evidence in the introductory paragraph of this section (titled, 'Some Crystallizing Wactives May Kill *C. elegans* through Multiple Independent Mechanisms'). If we were to suggest that this group of molecules kill by only a crystal-independent mechanism, it would ignore the four lines of evidence presented earlier in the manuscript that supports the model that crystals are capable of killing *C. elegans* larvae. Hence, we are left with the conclusion that this group of molecules must be capable of killing the animal through two

distinct mechanisms since, with this class of compounds, crystal formation is eliminated in the *sms-5* mutant yet the animals still die in the presence of these compounds.

We have clarified this idea by modifying the following sentence as follows:

We changed, 'This suggests that while some crystal-forming wactives may kill only via crystal formation, others may kill via two mechanisms- one that is crystal-dependent and another that is crystal-independent.', to, '*Given that several lines of evidence indicate that crystals can kill larvae, these data suggests that some crystal forming wactives may kill through both crystal formation and through crystal-independent mechanisms.*'

Reviewer 1-Comment 8: Page 15: bacteria in the environment can incorporate cholesterol into their membranes, and this nutrient would almost certainly be otherwise associated with particles in the environment. What is the evidence that "free" cholesterol (fully dissolved) is available to this organism in its ingesta? This is not compatible with the hypothesis of the function of marginal cells.

Author's Response: *C. elegans* can survive on bacterial-free media but still requires cholesterol, indicating that cholesterol can be absorbed by the worm through routes that have nothing to do with the ingestion of bacteria. Here are references to the literature describing the growth in bacterial-free media:

Samuel, T. K., Sinclair, J. W., Pinter, K. L., & Hamza, I. 2014. Culturing *Caenorhabditis elegans* in axenic liquid media and creation of transgenic worms by microparticle bombardment. *Journal of visualized experiments : JoVE*, (90), e51796. doi:10.3791/51796

Nass, R., and I. Hamza, 2007. The nematode *C. elegans* as an animal model to explore toxicology in vivo: solid and axenic growth culture conditions and compound exposure parameters. *Curr. Protoc. Toxicol.* Chapter 1: Unit1.9.

Szewczyk, N. J., E. Kozak, and C. A. Conley, 2003. Chemically defined medium and *Caenorhabditis elegans*. *BMC Biotechnol.* 3: 19.

Houthoofd, K., B. P. Braeckman, I. Lenaerts, K. Brys, A. De Vreese et al., 2002. Axenic growth up-regulates mass-specific metabolic rate, stress resistance, and extends life span in *Caenorhabditis elegans*. *Exp. Gerontol.* 37: 1371–1378.

Reviewer 1-Comment 9: Page 16: "blockage of the alimentary tract": this concept was not introduced in the manuscript; either expand or delete.

Author's Response: We have deleted the clause.

Reviewer 1-Comment 10a: Minor concerns: "precious" is not the word I would use; "essential scarce" works better for me.

Author's Response: On page 3, we changed 'precious' to 'essential and scarce'. On page 4, we simply deleted 'precious'. On page 15, we changed 'precious' to 'scarce'.

Reviewer 1-Comment 10b: Please use concentration instead of dose in the context of these experiments.

Author's Response: On page 8, we changed the two references to 'dose' to 'concentration'.

Reviewer 1-Comment 10c: Page 8: "emanating" is not the word I would choose; perhaps "breaching"?

Author's Response: We changed, '...we also observe crystal-like objects emanating from the marginal cells.', to, "...we also observe crystal-like objects in the marginal cells.'

Reviewer 1-Comment 10d: "at least 169..."; were there more?

Author's Response: We removed, 'at least' from this sentence.

Reviewer 1-Comment 10e: Second to the last line on page 11: fewer instead of less.

Author's Response: We have made the change (thank you).

Reviewer #2 (Remarks to the Author):

Reviewer 2-Opening Remarks-a: This manuscript investigates a phenomenon that many (including myself) might have dismissed as uninteresting and trivial – the formation of crystals in the pharynx of *C. elegans* treated with hydrophobic small molecules – in order to uncover a novel mechanism by which worms filter important small molecules from their environment, such as cholesterol.

Author's Response: We knew of the phenomenon for years and did what most would have done- ignored it. However, the phenomenon piqued the interest of the first author (Kamal) who was inspired to investigate it in more detail. Thank you for recognizing his insight!

Reviewer 2-Opening Remarks-b: This work has important implications for small molecule screening in *C. elegans* because it shows that many of the compounds that may be identified in screens as being lethal to worms might work through a common mechanism of crystal formation, and in order to find compounds that work through other mechanisms, one can screen in an sms-5 background, which largely prevents crystal formation. I found the conclusions of this paper to be well-supported, and so I only have a few comments.

Author's Response: Thank you for the kind words and for recognizing the value of the findings.

Reviewer 2-Comment 1a: It would be useful for the readership to know how common these crystal forming compounds are. The authors found that about 30% of the 238 wactives in Fig. 1 gave crystals or spheres, but perhaps these high percentages have something to do with the particular choice of the 238 wactives that were screened.

Author's Response: We believe the reviewer is correct. The 238 wactives are a highly biased set of molecules. Most of the 238 molecules that we surveyed are those that induce robust phenotypes at a concentration of at 60 μ M or less. And the wactive library itself is highly biased set of molecules: The library is a collection of 627 molecules that demonstrated some activity at a screening concentration of 60 μ M from our screen of 67012 distinct small molecules (see Burns et al., 2015).

To address this point, we have added the following sentences to the first paragraph of the results section: *'The wactive library is a collection of 627 compounds that we previously found to have bioactivity in C. elegans [14]. The 238 compounds that we surveyed are from plates 1 and 2 from the wactive library, which are enriched for potent bioactives, plus additional wactives that our group had working stocks of during the survey.'*

We have also modified a sentence in the abstract to recognize the fact (here, we included the term 'surveyed': *'...we found that more than one third of wactives surveyed visibly accumulate inside of the marginal cells as crystals or globular spheres.'*

We would like to note that one important point of the manuscript is not what fraction of randomly selected molecules form objects in the pharynx, but instead that drug-screener should be aware that molecules that are found to be lethal may be exerting their lethal effects through crystal formation.

Reviewer 2-Comment 1b: In particular, it appears that the percentage of the 238 wactives in Fig. 1 that kill the worms is higher than the percentage of the 508 wactives in Fig. 6.

Author's Response: Again, the reviewer is correct. The first two plates of the wactive library have more potentially lethal molecules than other plates in the library.

Reviewer 2-Comment 1c: Perhaps in Fig. 1, it would be more relevant to talk about the percentage of the wactives that kill worms that also give rise to crystals or spheres?

Author's Response: We thank the reviewer for this suggestion. We have added the following sentence to the first section: '*Of the 115 molecules that kill worms at a concentration of 30 μ M, 66 (57%) form crystals or spheres in at least 25% of the animals (Supplemental Dataset 1).*'

Reviewer 2-Comment 2: For the sms-5::flag-mcherry integrated reporter, it is referred to several names, including aMCp::sms-5::rfp and sms-5::flag-mcherry and sms-5(MC). I think these are all the same? It would be less confusing if one name were used throughout.

Author's Response: Agreed. We have added the sentence, 'For simplicity, we refer to this transgenic fusion protein as 'SMS-5(MC)' henceforth.' upon its introduction and use this notation throughout the manuscript.

Reviewer 2-Comment 3: The authors should show that the sms-5::flag-mcherry integrated reporter is specifically expressed in the anterior marginal cells. The image in SI Fig. 8 is too low resolution to tell.

Author's Response: We have updated Supplemental Figure 7 (which was Supplemental Figure 5 in the original submission) to show marginal cell specific expression of the SMS-5::FLAG::mCherry fusion protein in panels 'F' and 'G'.

Reviewer 2-Comment 4: Page 28 – For the molecular biology section, the authors say that detailed construct plans are available on request. However, the authors do not have that many constructs and so should easily be able to include these plans in the methods.

Author's Response: We have included the construct plans in the revised methods section. For convenience, we have pasted the modified section here:

pPRHM1051 is the wrm0626dC03 fosmid, which contains the *sms-5* locus (W07E6.3), wherein the SMS-5 coding sequence is C-terminally tagged with YFP. This construct rescues mutant *sms-5*'s resistance to wact-190. pPRHM1051 was created using Oliver Hobert's fosmid tagging methodology {Tursun, 2009 #4441}. Briefly, we PCR amplified the YFPint-FRT-galK-FRT cassette from the pBALU2 plasmid (a gift from Oliver Hobert) using the cassette-*sms-5* fusion primers, Fw (5'-cgagttccaaaacgtgtcgacattgaaaaaacacgaagatctttcgaatgagtaaaggagaagaactttcac-3') and Rev (5'-gagatttttattcaattttgttagcaaaaataattgttcagcctaattTAttgtatagttcatccatgcatg-3'). We transformed the bacterial strain carrying the wrm0626dC03 fosmid (a gift from Don Moerman) with the amplified PCR product and followed the detailed recombineering protocol described in Tursun *et al.*, 2009 to integrate the YFP tag directly into the *sms-5*-containing fosmid {Tursun, 2009 #4441}.

pPRHZ1138 (pgp-14p::SMS-5B(genomic)::FLAG::mCherry) (aka SMS-5(MC)). This construct is based upon pPRHM1065 (myo-2p::SMS-5::FLAG::mCherry). Briefly, we used primers HZ4pgp14hind.for (5'-aaattaagcttcaacagagagcaaggtg-3') and HZ3pgp14prokpn.rev (5'-aaattggtaccgttaattatcgatcatcg-3') to amplify by PCR a 1.6kb *pgp-14* promoter fragment from the pPRZH1160 (the *pgp-14* fosmid (WRM065dH09) 7.748kb Nael/SacI fragment in pKS) construct. We then cut the 1.6kb PCR fragment with HindIII and KpnI. We cut out the *myo-2* promoter in pPRHM1065 by using HindIII and KpnI and purifying the 7.5 kb fragment and ligating it to the 1.6 kb *pgp-14* promoter. We verified the construct through restriction digest analyses.

Reviewer 2-Comment 5: For SI Fig. 2, the arrows should be green to be consistent with other figures.

Author's Response: We have revised the figure as requested.

Reviewer 2-Comment 6a: For SI Fig. 3, how do the authors know that they are imaging the marginal cells as opposed to the intervening muscle cells? In figure A", is the tissue above and below the central lumen supposed to be marginal cells?

Author's Response: David Hall, our collaborator who did the TEM analyses along with his assistant Ken Nguyen, is the preeminent anatomist and electron microscopist in the *C. elegans* field. Of note, David Hall, along with Zeynep Altun, wrote the book on *C. elegans* anatomy (WormAtlas). In response to this question, David Hall writes, 'In our TEM analysis, most or all crystals are observed to lie at cell-cell borders in the apical zone of the anterior pharynx. In this region, the pharyngeal cells always meet in non-homologous pairings, as muscle cell-marginal cell appositions. Thus all crystals lie close to both cell types, but are observed to invaginate into the marginal cell rather than the muscle cell. Internal features (actinomyosin bundles vs intermediate filament bundles) help to some regard, but marginal cells always lie at the apex of narrow channels lining the lumen, while muscle cells make up the middle of the luminal wall. So, this geometry conclusively indicates that the crystal-type objects exist in the marginal cells and not the muscle cells.'

Reviewer 2-Comment 6b: It would be useful to have a figure to help orient the reader.

Author's Response: We have revised SI Fig. 3 (and made corresponding changes to the legend) to include a schematic to help orient the reader.

Reviewer 2-Comment 7: In SI Fig. 5 B', C', D', and E', the images should be rotated clockwise 90 degrees to make the orientation more consistent with the images above.

Author's Response: In order to keep the orientation of the worm heads in SI Fig 5 in the same orientation to all of the other worm head pictures in the manuscript, we have instead tilted the cartoons counter-clockwise ~90 degrees. In addition, we have added the same coloured dashed line to the micrograph that runs through the cartoon to help orient the reader to the spatial orientation of the cross-sections.

Reviewer 2-Comment 8: In SI Fig. 6, the labeling could be made more clear by including bars above and to the side of the images with grouped labels for the different types of images. The results described within the images should be placed just below the images to make the text more legible.

Author's Response: We have made the requested changes (thank you- it's much clearer this way).

Reviewer #3 (Remarks to the Author):

Reviewer 3-Opening Remarks: The authors describe the accumulation of some worm active substances (wactives) in the marginal cells of the pharynx. It is concluded that these are more hydrophobic substances from the pool. A genetic screen identified the sphingomyelin pathway to be involved in the resistance/accumulation to wactive-substances. In particular, SMS-5 (sphingomyelin syntase-5) is involved in wactive accumulation. The authors also show that SMS-5 might be responsible for cholesterol uptake.

Reviewer 3-Comment 1a: The experiments presented are performed thoroughly with all the controls required.

Author's Response: We thank the reviewer for the compliment.

Reviewer 3-Comment 1b: The whole study, however, is purely descriptive...

Author's Response: We think that there is great value in a description of previously uncharacterized phenomena that is of consequence to the growing field of groups using *C. elegans* as a drug-screening platform. That said, we respectfully argue against the idea that our work is purely descriptive. Our manuscript reports several experiments that test multiple hypotheses, some of which are summarized here:

- We tested the hypothesis that the crystalline objects might contribute to the death/arrest of the animal through the perforation of the plasma-membrane. When membrane-impermeable dye was used to test this hypothesis, we found that crystals do indeed damage the membrane to allow normally membrane impermeable dye to entre cells that form crystals.
- We tested the hypothesis that there are gene products that specifically promote object accumulation in the pharynx. Through our forward genetic screen, we do indeed find several genes that are specifically required for object accumulation.
- We tested the hypotheses that SMS-5 expression specifically in the cells that exhibit crystal formation in wild type animals could rescue: i) the *sms-5* mutant's resistance to crystal formation (Fig 4d); ii) the *sms-5* mutant's resistance to wact-190- and wact-416-induced death/arrest (Figs 4d and 6b and 6c); iii) the *sms-5* mutant's resistance to detergents (Fig 5a and 5b); and iv) the *sms-5* mutant's sensitivity to limited cholesterol (Fig 8). In every case, we found that marginal cell-expression was indeed sufficient to rescue these phenotypes.
- In response to another reviewer's comment, we tested whether molecules with the same physicochemical properties of the crystalizing compounds enrich for crystallizers from a newly purchased set of molecules and found that they do.
- In response to another reviewer's comment, we tested the hypothesis that small molecule precipitation in the media is sufficient to seed crystal formation in the marginal cells and found that it is not.

Reviewer 3-Comment 1c: ...giving no clue how sphingomyelin can be involved in the accumulation of wactives. Even the association of cholesterol uptake with SMS-5 activity does not provide any insight into the molecular mechanism of the process.

Author's Response: The reviewer is correct that in our original manuscript, we provided little mechanistic insight into how sphingomyelin-rich marginal cells act as a sink for hydrophobic

molecules. In fact, we were/are upfront about this in the discussion, where we write, ‘Exactly how SMS-5 facilitates the absorption of hydrophobic molecules is not clear.’ However, we neglected to cite published works that show that sphingolipids can act as a sink for cholesterol, and that the relationship is bi-directional in that cholesterol can trap sphingolipids. We have therefore modified the relevant section in the discussion to include the following sentence: *‘In addition, sphingolipids and cholesterol have been shown to act as molecular traps for one another [33, 34], which is consistent with the model that the SM-rich marginal cells of C. elegans can act as a sink for cholesterol and perhaps other hydrophobic molecules.’*

For the reviewer’s convenience- here are the relevant references:

Puri V., Jefferson J.R., Singh R.D., Wheatley C.L., Marks D.L., and Pagano R.E. 2003. Sphingolipid storage induces accumulation of intracellular cholesterol by stimulating SREBP-1 cleavage. *J Biol Chem.* 278(23):20961-70.

Eggeling C., Ringemann C., Medda R., Schwarzmann G., Sandhoff K., Polyakova S., Belov V.N., Hein B., von Middendorff C., Schönle A., and Hell S.W.. 2009. Direct observation of the nanoscale dynamics of membrane lipids in a living cell. *Nature.* 457(7233):1159-62. doi: 10.1038/nature07596.

It is important to note that while the manuscript provides no new mechanistic insight into how sphingomyelin acts as sink for hydrophobic small molecules, the manuscript does provide unique biological insight that uncovers a previously unknown role for the marginal cells of the pharynx, which has been a mystery in the *C. elegans* field for over three decades (when John White and colleagues first published TEM of serial sections of the worm in the mid 1980s).

Reviewer 3-Comment 1d: In addition, there is whole a zoology of wactive molecule classification into crystalizing and non-crystalizing ones.

Author's Response: We agree with the reviewer's comment. We reviewed the terminology use throughout and were able to eliminate 'class I crystalizing wactive' and 'class II crystalizing wactive', to reduce the number of notable categories to just four ('crystal-forming', 'sphere-forming', 'resistant', and 'hypersensitive'). We also hope that our new summary figure (Figure 9) helps to clarify a number of points. For convenience, we have pasted the summary figure here:

REVIEWERS' COMMENTS:

Reviewer #1 (Remarks to the Author):

The authors have clarified areas of the manuscript that had caused concern, both by refining the language and by adding additional data from new experiments. The explanations offered and revisions made resolve all my initial questions, and I recommend that the manuscript now be accepted for publication.

Reviewer #2 (Remarks to the Author):

The authors have adequately responded to all of my suggestions, except I would suggest that they incorporate the information in their response to Reviewer #2 - comment 6a into the manuscript in the figure legend or elsewhere so that the readers know how the marginal cells were identified.